# Reasoning Elicitation in Language Models via Counterfactual Feedback

**Alihan Hüyük,**[*†] **Xinnuo Xu,**[‡] **Jacqueline Maasch,**[§] **Aditya V. Nori,**[‡] **Javier González**[‡]
[†]Harvard University, [‡]Microsoft Research Cambridge, [§]Cornell Tech

## Abstract

Despite the increasing effectiveness of language models, their reasoning capabilities remain underdeveloped. In particular, causal reasoning through counterfactual question answering is lacking. This work aims to bridge this gap. We first derive novel metrics that balance accuracy in factual and counterfactual questions, capturing a more complete view of the reasoning abilities of language models than traditional factual-only based metrics. Second, we propose several fine-tuning approaches that aim to elicit better reasoning mechanisms, in the sense of the proposed metrics. Finally, we evaluate the performance of the fine-tuned language models in a variety of realistic scenarios. In particular, we investigate to what extent our fine-tuning approaches systemically achieve better generalization with respect to the base models in several problems that require, among others, inductive and deductive reasoning capabilities.

## 1 Introduction

Large language models (LLMs) are shown to be capable of delivering astounding performance in numerous tasks across various domains. Examples stretch from writing assistants (Gan et al., 2023), to sentiment analysis in social media (Simmering and Huoviala, 2023), and even applications in healthcare (González et al., 2023; Wong et al., 2023). While the ever-increasing accuracy of these systems is now undeniable, it is still rather unclear to what extent this accuracy is due to effective *recall* of their training data vs. a genuine ability to *reason* by extracting, understanding, and adapting the fundamental concepts underlying that training data (Huang and Chang, 2023; Li et al., 2023). Previous work suggests that LLMs might exhibit some emergent reasoning capabilities (Bubeck et al., 2023; Kıcıman et al., 2023). However, many have observed a significant *reasoning-recall gap*: LLMs still perform substantially better on recall-based tasks that do not explicitly require reasoning (Zhang et al., 2023a; Ahn et al., 2024; Seals and Shalin, 2024).

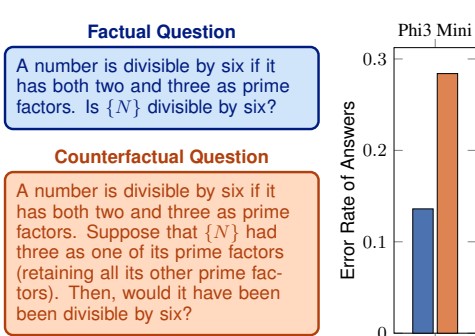

**Factual Question**

A number is divisible by six if it has both two and three as prime factors. Is $\{N\}$ divisible by six?

**Counterfactual Question**

A number is divisible by six if it has both two and three as prime factors. Suppose that $\{N\}$ had three as one of its prime factors (retaining all its other prime factors). Then, would it have been been divisible by six?

Figure 1: *Error rate of Phi3-Mini in answering factual vs. counterfactual questions*—sampling 10 answers for each $N \in \{1, \ldots, 100\}$. It performs disproportionately better for the factual question (cf. recall) as opposed to the counterfactual question (cf. reasoning).

Motivated by this discrepancy between how well LLMs can recall vs. reason, our goal in this paper is to see whether they can be fine-tuned explicitly to improve their reasoning. While reasoning can take different forms, we will focus on *causal reasoning* as it provides us with a clear distinction between recall and reasoning[1]: the former is limited to inferring statistical correlations, whereas the latter involves working with interventions and counterfactuals (Pearl, 2000). It has been previously shown that LLMs struggle with counterfactual questions compared to purely factual questions (Jin et al., 2024; González and Nori, 2024). This difficulty highlights the recall-reasoning discrepancy within the causal domain (Figure 1).

---

[*]Work done as an intern at Microsoft Research Cambridge. Correspondence: `ahuyuk@seas.harvard.edu`

[1]As an example, a different kind of reasoning would be *symbolic reasoning*, which involves manipulating symbols that represent mathematical statements (MacColl, 1897; Kelley, 1992).

Adopting a causal framework also allows us to consider an LLM's ability to identify higher concepts that are essential for connecting causes to their effects in causal reasoning, such as *necessity* and *sufficiency* (Mackie, 1965; Lewis, 1973; Halpern and Pearl, 2005). For instance, a cause $X$ is said to be necessary for an effect $Y$ if (i) without intervention, $X$ and $Y$ occur together and (ii) intervening to remove $X$ results in no $Y$ (Pearl, 1999). Therefore, for an LLM to be able to identify that $X$ is necessary for $Y$, it needs to not only determine the factual in (i) is indeed the case but also simultaneously recognize the counterfactual would have been different as in (ii). This makes identification of necessity, or similar relationships like sufficiency, a particularly good test of reasoning because it requires the LLM to understand when to recall (cf. factual thinking) vs. when to reason (cf. counterfactual thinking).

We improve the causal reasoning of LLMs by adapting established methods of fine-tuning. In particular, we consider *supervised fine-tuning* (SFT, e.g. Dai and Le (2015); Peters et al. (2018); Radford et al. (2018); Khandelwal et al. (2019); Howard and Ruder (2018), used in Ziegler et al. (2019); Ouyang et al. (2022)) and *direct preference optimization* (DPO, Rafailov et al. (2024), used in Tian et al. (2023); Lin et al. (2024a)). For both of these approaches, we propose procedures to generate supervised and preference-based datasets using factual questions as well as counterfactual questions. We argue that generating demonstrations on a question-by-question basis only improves the correctness of individual answers. As we discussed, identifying higher concepts such as necessity and sufficiency requires coordination between how factual and counterfactual questions are answered together. To target these higher concepts directly, we propose generating preference-based datasets over dialogues involving both factual and counterfactual questions.

When the goal of fine-tuning is specifically to improve reasoning, a unique problem arises in evaluating the fine-tuned LLMs: we cannot just measure performance for a held-out set of test samples within the same reasoning task. If we do, it would be impossible to tell whether the LLM actually learned to reason or whether it is still recalling the demonstrations we have made during fine-tuning.[2] Hence, measuring the generalization performance with respect to new reasoning tasks becomes crucial.[3] We cannot expect fine-tuning on one problem instance to arbitrarily generalize to all problem instances either. So, building a systematic understanding regarding to what extent fine-tuning for reasoning should be expected to generalize becomes important as well.

To build that understanding, we identify different modes in which reasoning in one problem is transferred to other problems. Notably, we define *inductive generalization* and *deductive generalization*. Given a causal system where $X \to Y \to Z$, inductive generalization is the ability to reason about the transitive relationship $X \to Z$ when demonstrated how to reason about $X \to Y$ and $Y \to Z$. Conversely, deductive generalization is the ability to reason about the relationships $X \to Y$ and $Y \to Z$ when demonstrated how to reason about $X \to Z$. We show that fine-tuning for reasoning generalizes much more effectively in an inductive mode rather than a deductive mode (among many other insights in Section 5).

**Contributions.** We have four major contributions, corresponding to each of the following sections:

**§2** We describe a framework for fine-tuning based on causal reasoning and formally categorize the ways in which reasoning generalizes from one problem to another. These categories are *common-effect*, *common-cause*, *inductive*, and *deductive*.

**§3** We introduce novel metrics to measure the reasoning performance of an LLM, defining *necessity and sufficiency inconsistency rates* (N-IR & S-IR) based on probabilities of necessity and sufficiency from the causality literature. We also introduce the concepts of *absent necessity* and *absent sufficiency* to supplement cause-effect relationships covered neither by necessity nor sufficiency.

**§4** We propose procedures to generate datasets to be used with SFT and DPO to fine-tune for reasoning by incorporating *counterfactual feedback*. In particular, we argue for generating dialogues that involve paired factual and counterfactual questions to directly target the reasoning metrics we introduce in Section 3. We call this *causal consistency feedback*.

**§5** Finally, we evaluate the performance of the procedures proposed in Section 4 using the metrics introduced in Section 3. Moreover, we investigate to what extent that performance generalizes in relation to our categorization in Section 2.

---

[2]For instance, chain-of-thought prompting aims to improve reasoning by providing examples of how a problem can be solved in smaller steps. While such prompting is effective, unless tested on cases that require a novel rearrangement of those smaller steps, its effectiveness can be attributed to successful imitation of the provided examples and is not necessarily the result of true reasoning (Wei et al., 2022, see Appendix E.1 for further discussion).

[3]We are interested in this particular notion of generalization although there are other notions, see Appendix F.

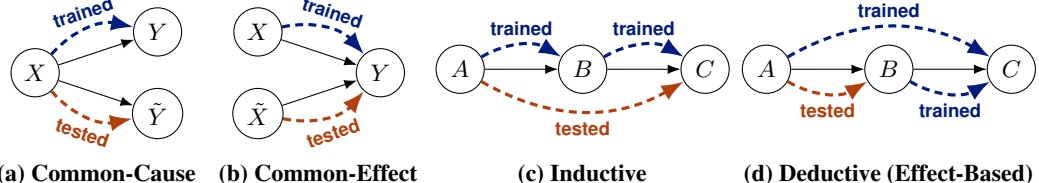

(a) Common-Cause      (b) Common-Effect      (c) Inductive      (d) Deductive (Effect-Based)

Figure 2: *Different modes of generalization,* in terms of the cause-effect relationships demonstrated during fine-tuning (i.e. $\mathcal{D}$, **blue**) vs. the relationship that the fine-tuned model is evaluated on (i.e. $\mathcal{P}_{X \to Y}$, **orange**).

## 2 FINE-TUNING FOR REASONING

**World Model.** We consider a causal world model, in which $X$ (cause) and $Y$ (effect) are two binary variables, indicating the absence or presence of some conditions. We will denote with $x, y$ the values taken by $X$ and $Y$ respectively when the conditions they represent are present, and with $x', y'$ the complements of these values (i.e. the values taken by $X$ and $Y$ when the conditions they represent are absent). The *context*, denoted by $U$, consists of all exogenous variables. Without any loss of generality, we assume that all randomness in the model is captured through these exogenous variables, and all endogenous variables, including $X$ and $Y$, are deterministic functions of the exogenous variables (i.e. the context $U$). We denote these deterministic functions as $X = f_X(U)$ and $Y = f_Y(X, U)$. Additionally, we denote the the *potential effects* under the potential interventions for each unit in the population as $Y_x = Y|do(X = x) = f_Y(x, U)$ and $Y_{x'} = Y|do(X = x') = f_Y(x', U)$.

**Language Model.** We can estimate different effects using a language model. Formally, let $q(u)$ be a *factual question template* that describes the world model in natural language and asks what the factual effect would be for a specific context $u$. Denoting the language model by $\ell$, let $a = \ell(q(u))$ be the model's answer to this question, which will be in natural language form. To transform the answer into binary form, we use a mapping $h$ such that $\hat{Y} = h(a) = h(\ell(q(u))) \in \{y, y'\}$.[4] Similar to the factual case, suppose we also have *interventional question templates* $\tilde{q}_x(u)$ and $\tilde{q}_{x'}(u)$ that describe the world model. However, these templates ask for the potential effects under interventions $do(X = x)$ or $do(X = x')$. This leaves us with the following estimates for the two potential effects. For a given context $u$, we rely on the factual question template when the effect is factual, and on the interventional question template when the effect is counterfactual:

$$\hat{Y}_x = \begin{cases} h \circ \ell \circ q(U) & \text{if } X = x \\ h \circ \ell \circ \tilde{q}_{x'}(U) & \text{if } X = x' \end{cases} \qquad \hat{Y}_{x'} = \begin{cases} h \circ \ell \circ \tilde{q}_x(U) & \text{if } X = x \\ h \circ \ell \circ q(U) & \text{if } X = x' \end{cases} \qquad (1)$$

**Problem.** Let $\mathcal{P}$ describe the context distribution such that $U \sim \mathcal{P}$. Moreover, let $\mathcal{P}_{X \to Y}$ denote the corresponding distribution of cause $X = f_X(U)$ and potential effects $Y_x(U) = f_Y(x, U)$, $Y_{x'}(U) = f_Y(x', U)$ such that $X, Y_x, Y_{x'} \sim \mathcal{P}_{X \to Y}$. Suppose we are interested in optimizing some metric $\mathbb{V}[\ell; \mathcal{P}_{X \to Y}] \in \mathbb{R}$ that measures the reasoning performance of the language model $\ell$ for the cause-effect relationship $\mathcal{P}_{X \to Y}$ (we discuss the design of $\mathbb{V}$ in the subsequent section). Then, the problem of *fine-tuning for reasoning* can be expressed as

$$\text{maximize} \quad \mathbb{V}[\ell; \mathcal{P}_{X \to Y}] \quad \text{given} \quad \ell_0, \mathcal{P}, \mathcal{D} = \{\mathcal{P}_{X_i \to Y_i}\}_i \qquad (2)$$

where $\ell_0$ is the *target language model*, and $\mathcal{D}$ is the set of different cause-effect relationships $\mathcal{P}_{X_i \to Y_i}$ that are available as *demonstrations*. These relationships may involve causes $\{X_i\}$ and effects $\{Y_i\}$ other than the cause $X$ or the effect $Y$ of interest. We refer to the case where only the cause-effect relationship of interest is demonstrated such that $\mathcal{D} = \{\mathcal{P}_{X \to Y}\}$ as the "**in-domain**" problem.

**Modes of Generalization.** As we have discussed in the introduction, an in-domain evaluation is not sufficient alone to assess the success of fine-tuning for reasoning. Therefore, we categorize different ways in which reasoning can generalize—that is, how $\mathcal{D}$ might relate to $\mathcal{P}_{X \to Y}$ when $\mathcal{P}_{X \to Y} \notin \mathcal{D}$. We identify four main structures, summarized in Figure 2:

(i) **Common-Cause:** When the relationship $X \to Y$ is demonstrated, *common-cause generalization* refers to the ability to reason about other relationships $X \to \tilde{Y}$ that involve the same cause $X$.

---

[4]In practice, this mapping would also be a language model prompted to reduce given answers to a binary "negative" or "positive" (see the appendix for the exact prompt used in this paper).

(ii) **Common-Effect:** When the relationship $X \to Y$ is demonstrated, *common-effect generalization* refers to the ability to reason about other relationships $\tilde{X} \to Y$ that involve the same effect $Y$. Unlike common-cause generalisation, the task of determining the factual effect without intervention remains the same, regardless of whether $X$ or $\tilde{X}$ is the cause of interest.

(iii) **Inductive:** When the relationship $A \to B$ and $B \to C$ are demonstrated, *inductive generalization* refers to the ability to reason about the transitive relationship $A \to C$. This ability may be hindered if $A$ has a direct effect on $C$ that is not mediated by $B$. We will investigate this empirically in Section 5.

(iv) **Deductive:** Similar to inductive generalization, consider the causal relationship $A \to B \to C$. When the relationships $A \to C$ and $B \to C$ are demonstrated, *effect-based deductive generalization* is the ability to reason about the relationship $A \to B$. Similarly, when the relationships $A \to C$ and $A \to B$ are demonstrated, *cause-based deductive generalization* refers to the ability to reason about the relationship $B \to C$. We will investigate the potential differences between the two scenarios empirically in Section 5.

## 3 METRICS OF REASONING

Having defined the problem of fine-tuning for reasoning, we now discuss what would be a good measure of reasoning ability (i.e. a good choice for $\mathbb{V}$). In Section 3.1, we define error rates based on the *correctness* of answers given by the language model to individual questions. In Section 3.2, we go beyond these simple error rates and propose various inconsistency rates that capture the *causal consistency* between the factual and counterfactual answers given within the same context. As emphasized in the introduction, such consistency is necessary to identify causal relationships such as necessity and sufficiency. Later, in Section 4, we will describe various methods for generating datasets that aim to optimize either of these metrics.

### 3.1 CORRECTNESS

Ignoring relationship between factual and counterfactual effects, the correctness of an individual answer $a = \ell \circ q(u) \mid \ell \circ \tilde{q}_x(u) \mid \ell \circ \tilde{q}_{x'}(u)$ can be characterized by the *factual error rate* (F-ER) and the *counterfactual error rate* (CF-ER) respectively:

$$\text{F-ER} = \mathbb{P}\{\hat{Y} \neq Y\} \qquad \text{CF-ER} = \mathbb{P}\left\{ \begin{array}{ll} \hat{Y}_{x'} \neq Y_{x'} & \text{if } X = x \\ \hat{Y}_x \neq Y_x & \text{if } X = x' \end{array} \right\} \qquad (3)$$

where $\hat{Y}$, $\hat{Y}_x$, and $\hat{Y}_{x'}$ represent the binary values implied by the answer $a$. Using these two metrics, we define the *average error rate* as Avg-ER $= (\text{F-ER} + \text{CF-ER})/2$.

**Why are factual and counterfactual correctness alone not enough?** Being able to correctly estimate factuals (cf. F-ER) or counterfactuals (cf. CF-ER) is, of course, an important step in causal reasoning. However, what we ultimately want is to characterize the relationship between a cause and its effect. For instance, is the cause necessary for the effect to occur? Is it sufficient? Or do the cause and the effect only occur together (necessary and sufficient)? Identifying such relationships rely on the estimated factuals and counterfactuals collectively—only getting one right but not the other might not always lead to a correct characterization of the cause-effect relationship. By measuring the factual and counterfactual accuracy separately, F-ER and CF-ER fail to capture any dependencies between the two answers and how they might be describing a larger relationship together.

As a concrete example, consider *necessity*. According to Pearl (1999), when a cause $X$ and an effect $Y$ occur together (i.e. $X = x$ and $Y = y$), the cause is said to have been necessary for the effect if the effect would not have occurred in the absence of the cause (i.e. $Y_{x'} = y'$). Making an accurate judgement regarding whether there is a necessity relationship between $X$ and $Y$ requires both $\hat{Y}$ and $\hat{Y}_{x'}$ to be correct when $X = x$ and $Y = y$. However, no factual or counterfactual estimate needs to be correct when $X = x'$ (as it is immediately apparent that cases where $X = x'$ do not affect necessity), and similarly, only the factual estimates needs to be correct when $X = x$ but $Y = y'$. F-ER and CF-ER do not account for this complex requirement at all. In particular, depending on how $X$ and $Y$ are distributed, a language model can achieve F-ER and CF-ER as high as $1/2$ by always estimating either $Y_x$ or $Y_{x'}$ correctly (but not both together) while never reaching an accurate conclusion regarding necessity.

## 3.2 CAUSAL CONSISTENCY

Previous work (González and Nori, 2024) has considered the use of "probabilities of causation" together with F-ER and CF-ER to provide a set metrics that fully characterize the relationship between a cause and its effect. Similar to necessity, Pearl (1999) also provides a causal definition of sufficiency: whether the cause would have produced the effect (i.e. $Y_x = y$) when both the cause and the effect are absent (i.e. $X = x'$ and $Y = y'$). The *probability of necessity* (PN) and the *probability of sufficiency* (PS) are defined as:

$$\text{PN} := \mathbb{P}\{Y_{x'} = y' | X = x, Y = y\} \qquad \text{PS} := \mathbb{P}\{Y_x = y | X = x', Y = y'\} \qquad (4)$$

The answers given by the language model to factual and counterfactual questions and the effects $\hat{Y}_x, \hat{Y}_{x'}$ estimated from those answers naturally induce an empirical pair of PN and PS values:

$$\widehat{\text{PN}} = \mathbb{P}\{\hat{Y}_{x'} = y' | X = x, \hat{Y} = y\} \qquad \widehat{\text{PS}} = \mathbb{P}\{\hat{Y}_x = y | X = x', \hat{Y} = x'\} \qquad (5)$$

**Why are PN and PS correctness alone not enough?** To evaluate reasoning in language models, González and Nori (2024) use (1) a probabilistic measure ($\gamma$-overlap) to assess how well the distributions of $\widehat{\text{PN}}$ and $\widehat{\text{PS}}$ match the true PN and PS, and (2) the factual and counterfactual error rates. We refine this approach by defining unifying metrics that simultaneously take both aspects of the problem into account, thereby simplifying the evaluation process.

Due to the averaging done by probabilities, achieving a perfect PN-PS with the language model only requires identifying correct vs. predicted marginal frequencies, without needing individual units to be accurate. Although this is captured by the factual and counterfactual error rates F-ER and CF-ER, it is convenient to have a single metric that encapsulates both dimensions of the problem. We address this by requiring the necessity or sufficiency relationships identified by the language model to be accurate on a unit-by-unit basis. A unit is a realization of the exogenous variable $U$. It induces the values of $X$ and $Y$ as well as the counterfactual outcome $Y_{X'}$, where $X'$ represents the complement of the observed $X$ regardless of its value. Note that $Y_X = Y$ is the factual outcome.

We focus on *necessity* where a unit/context might exhibit one of three situations: (i) Necessity occurs, denoted by "$\mathbb{N}$", meaning that both $X$ and $Y$ occur, $X = x$ and $Y = y$, and the cause was necessary for the effect, $Y_{X'} = y'$. (ii) Necessity does not occur, which we denote by "$\mathbb{N}'$", meaning that both $X$ and $Y$ occur but the cause was not necessary for the effect, $Y_{X'} \neq y'$. (iii) Not relevant case as necessity is concerned, which we denote by $\emptyset$, when neither $X$ nor $Y$ (or both) did occur. Since value of the context variable $U$ fully characterizes the unit, we can define unit-wise necessity as

$$\mathcal{N}(X, Y, Y_{X'}; U) = \begin{cases} \mathbb{N} & \text{if } X = x \wedge Y = y \wedge Y_{X'} = y' \\ \mathbb{N}' & \text{if } X = x \wedge Y = y \wedge Y_{X'} \neq y' \\ \emptyset & \text{if } X = x' \vee Y = y' \end{cases} \qquad (6)$$

The *necessity inconsistency rate* (N-IR) is the frequency with which the language model estimates the unit-wise necessity $\mathcal{N}$ inaccurately (see Appendix G for an alternative interpretation):

$$\text{N-IR} := \mathbb{E}_{\mathcal{P}(U)}[\mathcal{N}(X, \hat{Y}, \hat{Y}_{X'}; U) \neq \mathcal{N}(X, Y, Y_{X'}; U)], \qquad (7)$$

where $\mathbb{E}_{\mathcal{P}(U)}$ denotes the expectation over $U$ and $\hat{Y}, \hat{Y}_{X'}$ are the analogous factual and counterfactuals to $Y, Y_{X'}$ estimated from the model. Remark that $\text{PN} = \mathbb{E}_{\mathcal{P}(U)}[\mathcal{N} = \mathbb{N} | \mathcal{N} \neq \emptyset]$ by construction. Also note that N-IR $= 0$ implies that $\widehat{PN} = \text{PN}$. However, errors made in different units can no longer 'balance each other out' to achieve N-IR $= 0$. We can also define context-wise sufficiency $\mathcal{S}$ in an analogous way: (i) $\mathcal{S} = \mathbb{S}$ if $X = x', Y = y', Y_{X'} = y$, (ii) $\mathcal{S} = \mathbb{S}'$ if $X = x', Y = y', Y_{X'} \neq y$, and (iii) $\mathcal{S} = \emptyset$ otherwise. This induces the *sufficiency inconsistency rate* S-IR $= \mathbb{P}\{\hat{\mathcal{S}} \neq \mathcal{S}\}$.

Neither PN and PS nor the inconsistency rates N-IR and S-IR are sensitive to all answers given by the language model. This is because necessity and sufficiency only concern cases where $X = x, Y = y$ and $X = y', Y = y'$. For instance, when $X = x'$ and $Y = y$ and the factual effect has been estimated correctly such that $\hat{Y} = Y$, the counterfactual estimate $\hat{Y}_x$ has no impact on PN, PS, N-IR, or S-IR. Regardless of whether $\hat{Y}_x = Y_x$, all four quantities stay the same. To cover all possible counterfactuals we can ask a language model for, it makes sense to also evaluate counterfactuals of the type $Y_{x'} = y | X = x, Y = y'$ and $Y_x = y' | X = x', Y = y$. Of course, the probabilities of these counterfactuals can be defined by means of PN and PS by changing the default observed state. However, here we name them as *absent necessity* and *absent sufficiency* to be explicit about the

two extra cases where language model can make mistakes.[5] In our context-based framework, the corresponding context-wise $\mathcal{AN}$ and $\mathcal{AS}$ are defined in a similar fashion as $\mathcal{N}$ and $\mathcal{S}$, which induce the inconsistency rates AN-IR $= \mathbb{P}\{\hat{\mathcal{AN}} \neq \mathcal{AN}\}$ and AS-IR $= \mathbb{P}\{\hat{\mathcal{AS}} \neq \mathcal{AN}\}$. As a reasoning metric, we define the *average inconsistency rate* as Avg-IR $=$ (N-IR + S-IR + AN-IR + AS-IR)$/4$. This metric has the following properties: (i) it accounts for all characterizations of the necessity and sufficiency of the target causal effect, and (ii) it is unit-dependent, so factual and counterfactual accuracy errors cannot be balanced out.

**An Illustrative Example.** We illustrate the difference between correctness and causal consistency as follows. Consider the following language models: (i) *Factually Correct* answers all factual questions correctly (F-ER $= 0$) but makes occasional mistakes in answering counterfactual questions. This represents an extreme version of the imbalance highlighted in the introduction. (ii) *Uniformly Correct* makes both factual and counterfactual mistakes at equal rates (F-ER $=$ CF-ER), but these mistakes happen independently of each other. (iii) *Causally Consistent* reasons on a unit-by-unit basis (as opposed to question-by-

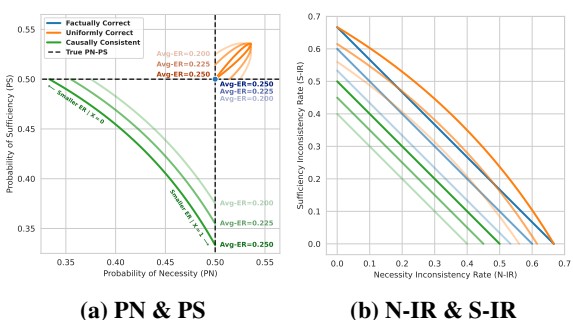

(a) PN & PS  (b) N-IR & S-IR

Figure 3: *Causal consistency vs. correctness.* Despite having the same Avg-ER, different types of error distributions lead to widely different PN & PS characteristics.

question) and either gets both the factual question and counterfactual question right or gets both of them wrong. Suppose the cause never prevents the effect such that $(X, Y_x, Y_{x'}) \in \{(x, y', y'), (x, y', y), (x, y, y), (x', y', y'), (x', y', y), (x', y, y)\}$ (with equal probabilities).

Figure 3 shows the PN & PS as well as N-IR & S-IR of these models for fixed levels of Avg-ER as the error shifts between contexts where the cause may be necessary (i.e. $X = x$) vs. contexts where it may be sufficient (i.e. $X = x'$). Despite having the same Avg-ER, the three models induce widely different PN & PS values, representing different causal interpretations. While Figure 3a might suggest that the factually correct models are the best performing, this is purely coincidental. Due to the averaging done by PN & PS, the mistakes made in different units end up balancing each other out. Looking at N-IR & S-IR in Figure 3b reveals that the causally consistent models are actually the best, even outperforming models with significantly smaller Avg-ER.

## 4 FINE-TUNING WITH COUNTERFACTUAL FEEDBACK

Despite the significant differences between correctness and causal consistency, success in either metric relies on accurate estimates of counterfactual outcomes. Therefore, to solve the fine-tuning problem in (2), it is essential to leverage the counterfactual information available in demonstrations $\mathcal{D}$, irrespective of the metric we aim to target as $\mathbb{V}$. We present a data-centric approach to achieve this and propose three methods for generating datasets using counterfactual feedback. These datasets can then be utilised by existing algorithms for fine-tuning such as SFT or DPO. These methods are summarized in Figure 4.

**Supervised Counterfactual Feedback.** Recall that we assumed access to an extractor $h$ that can reduce answers given in natural language to binary outcomes $\hat{y} = h(a) \in \{y, y'\}$. Now, further suppose that we can perform this extraction in reverse, denoted as $H$: Given a question $q$ and the true outcome $y_{\text{true}}$ corresponding to this question, we can form a natural language answer $a = H(q, y_{\text{true}})$. In practice, we achieve this by prompting a language model to provide an answer to question $q$ that starts with *"Yes"* or *"No"* (see the appendix for the full prompt). Based on these answers, we generate a dataset $\mathbb{D}$ of both factual and counterfactual questions and their answers:

$$\mathbb{D} = \{ \; q_f = q(U), \quad a_f = H(q_f, Y), \\ q_{cf} = \tilde{q}_{X'}(U), \; a_{cf} = H(q_{cf}, Y_{X'}) \; \}_{U, X, Y, Y_{X'} \sim \mathcal{D}}$$

This dataset can directly be used with any SFT algorithm to fine-tune the target model $\ell_0$.

---

[5]Note that the use of these quantities is an alternative but equivalent characterization of all possible counterfactual outcomes to the one in Pearl (1999), where the probabilities of disablement $Y_{x'} = y'|Y = y$ and the probability of enablement $Y_x = y|Y = y'$ are introduced.

**Preference-based Counterfactual Feedback.** SFT can be limited by the quality of answers generated as ground-truth and their similarity to the model's original answers. Without access to a language model that is already better at reasoning than our target model, it might challenging to build an answer generator $H$ that provides high quality samples. In that case, it is desirable to provide direct feedback to the answers generated by the target language model. We do so by first generating multiple answers to different questions (using a high sampling temperature to get sufficient variation):

$$\mathbb{D} = \{ \; U, \; q_f = q(U), \quad a_f[1] \sim \ell_0(q_f), \; \ldots, \; a_f[N] \sim \ell_0(q_f), $$
$$q_{cf} = \tilde{q}_{X'}(U), \; a_{cf}[1] \sim \ell_0(q_{cf}), \; \ldots, \; a_{cf}[N] \sim \ell_0(q_{cf}) \; \}_{U,X,Y,Y_{X'} \sim \mathcal{D}} \quad (8)$$

Then, we form a preference-based dataset where correct answers are preferred over incorrect answers:

$$a_f[i] \succ a_f[j] \quad \Longleftrightarrow \quad \mathbb{1}\{h(a_f[i]) = Y\} \quad > \quad \mathbb{1}\{h(a_f[j]) = Y\} \quad (9)$$
$$a_{cf}[i] \succ a_{cf}[j] \quad \Longleftrightarrow \quad \mathbb{1}\{h(a_{cf}[i]) = Y_{X'}\} \quad > \quad \mathbb{1}\{h(a_{cf}[j]) = Y_{X'}\} \quad (10)$$

The DPO algorithm can directly be used with this dataset to maximize the likelihood of preferred answers (i.e. $a[i]$) relative to the answers they are preferred over (i.e. $a[j]$).

**Preference-based Causal Consistency Feedback.** Running DPO with preferences determined by a reward function, where alternatives with higher rewards are preferred over those with lower rewards, is equivalent to maximizing that reward function (Rafailov et al., 2024). In our case, this means that running DPO with the above preferences would, in effect, minimize the average error rate (i.e. Avg-ER), as these preferences are generated by treating correctness (i.e. $\mathbb{1}\{h(a) = \hat{Y} = Y\}$) as a reward function. To target the inconsistency rates introduced in Section 3.2, we propose to (i) pair factual and counterfactual questions, (ii) prompt the target language model to answer them simultaneously, and then (iii) elicit preferences based on the joint answer. Formally,

$$(a_f[i], a_{cf}[i]) \succ (a_f[j], a_{cf}[j]) \quad \Longleftrightarrow \quad \mathcal{R}(h(a_f[i]), h(a_{cf}[i]); U) > \mathcal{R}(h(a_f[j]), h(a_{cf}[j]); U) \quad (11)$$

where $\mathcal{R}(\hat{Y}, \hat{Y}_{X'}; U) = \mathbb{1}\{\mathcal{N} = \hat{\mathcal{N}}\} + \mathbb{1}\{\mathcal{S} = \hat{\mathcal{S}}\}$ $+ \mathbb{1}\{\mathcal{AN} = \hat{\mathcal{AN}}\} + \mathbb{1}\{\mathcal{AS} = \hat{\mathcal{AS}}\}$. We call this *causal consistency feedback* (CCF) (see Figure 4b vs. 4c). CFF explicitly targets Avg-IR rather than Avg-ER and can still be used directly with the DPO algorithm.

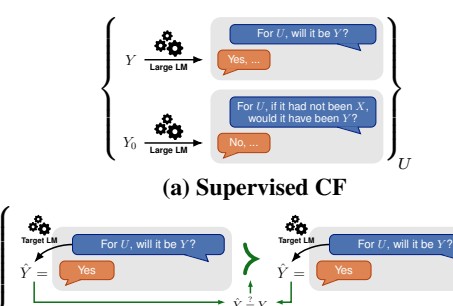

**(a) Supervised CF**

## 5 EXPERIMENTS

We begin with a proof-of-concept case study. We analyze a hand-crafted puzzle to assess the effectiveness of all fine-tuning techniques introduced in Section 4 when trained on different types of datasets within the context of the in-domain causal reasoning scenarios (**§5.1**). We also address the research question posed in Section 1, *i.e.*, to what extent the performance improvements in causal reasoning achieved through the fine-tuning process generalize across all the generalization modes (**§5.2**). Subsequently, we use three additional real-world problems to examine our findings (**§5.3**).

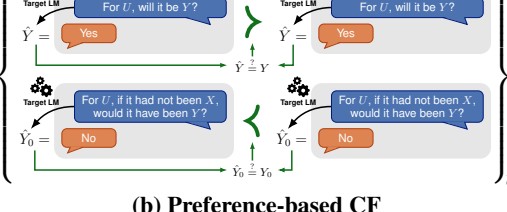

**(b) Preference-based CF**

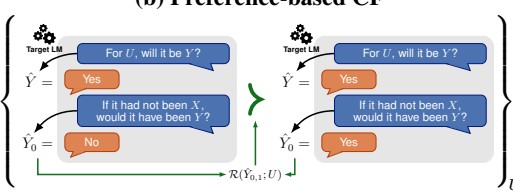

**(c) Preference-based CCF**

### 5.1 IN-DOMAIN REASONING

We evaluate all fine-tuning techniques when trained on various types of demonstrations in a synthetic in-domain reasoning problem.

**Experimental Setup.** See Figure 5. The puzzle describes a candy party. The context is defined by the four-dimensional random vector $U = (N_A, N_B, N_C, N_D)$ where each element follows the same uniform distribution $\mathcal{U}(1, 12)$. The causal structure, derived from the narratives,

Figure 4: *Summary of the proposed fine tuning methods.* Supervised and preference-based counterfactual feedback (CF) target correctness: the former by generating correct answers given each question and the latter by sampling answers and preferring the correct ones over the others. Causal consistency feedback (CCF) targets causal consistency instead: Asking both the factual and the counterfactual questions within the same dialogue allows us to elicit preferences according to relationships between the factual and counterfactual answers.

is presented in the middle section of the figure. We selected *A: Anna is happy or not* as the cause ($X$), and *D: Dave is happy or not* as the effect ($Y$). The factual questions $q(u)$ are obtained by randomly drawing values for the four numerical variables from the distribution $\mathcal{U}$, The counterfactual questions $\tilde{q}_{X'}(u)$ are generated by introducing an assumption that negates the cause *i.e.* if in the context $A$ is "*Anna is happy*" based on the value of $N_A$, the injected assumption would be "*suppose that Anna is not happy*", and vice versa. Since we are assessing the in-domain reasoning scenario, the cause-effect demonstration used during the fine-tuning phase are likewise employed in the evaluation phase.

We first generate dataset $\mathbb{D} = \{(q(u), a_f)\} \cup \{(\tilde{q}_{X'}(u), a_{cf})\}$ for each fine-tuning techniques introduced in Section 4 following the algorithms shown in Appendix C. Then we fine-tune the mini version of Phi-3 (Abdin et al., 2024) on $\mathbb{D}$. We include five baselines: the base language model (Phi-3 mini) without fine-tuning (***Base***), the base model fine-tuned using the SFT and DPO methods on factual examples $\{(q(u), a_f)\}$ exclusively (***SFT-OnlyF*** and ***DPO-OnlyF***), and the base model fine-tuned using the SFT and DPO methods on counterfactual examples $\{(\tilde{q}_{X'}(u), a_{cf})\}$ exclusively (***SFT-OnlyCF*** and ***DPO-OnlyCF***). For OnlyF and OnlyCF, we have doubled the number of contexts sampled so that every method still has access to the same number of question-answer examples. As our proposed methods, we include the base model fine-tuned using SFT, DPO, and CCF methods on both factual and counterfactual examples (***SFT-F&CF***, ***DPO-F&CF***, and ***DPO+CCF***).

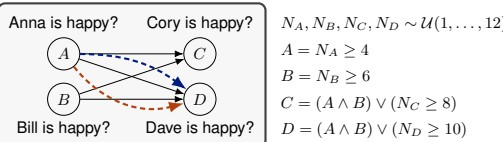

> **Question:** Anna, Bill, Cory, and Dave are going to a party, where the host is going to distribute candies. Anna will be happy if she gets at least 4 candies. Bill will be happy if he gets at least 6 candies. Cory will be happy if Anna and Bill are both happy or if he gets at least 8 candies. Dave will be happy if Anna and Bill are both happy or if he gets at least 10 candies. After distributing the candies, Anna gets $N_A$, Bill gets $N_B$, Cory gets $N_C$, and Dave gets $N_D$. Is Dave happy?

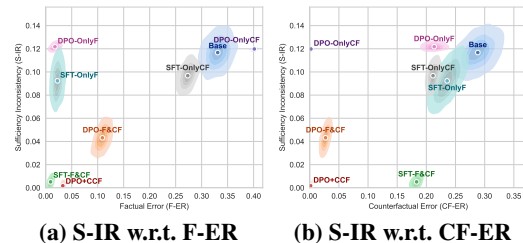

$$N_A, N_B, N_C, N_D \sim \mathcal{U}(1, \ldots, 12)$$
$$A = N_A \geq 4$$
$$B = N_B \geq 6$$
$$C = (A \wedge B) \vee (N_C \geq 8)$$
$$D = (A \wedge B) \vee (N_D \geq 10)$$

> **Counterfactual Question:** Now, suppose that Anna is (not) happy regardless of the candy distribution. With this assumption, is Dave happy?

Figure 5: *Hand-crafted puzzle* with the original factual question, causal model with structural equations, and a counterfactual question. **Blue** and **orange** arrows show the cause-effect interventions demonstrated to the model during fine-tuning and evaluation phases.

**(a) S-IR w.r.t. F-ER**   **(b) S-IR w.r.t. CF-ER**

Figure 6: *In-domain results in the candy party puzzle.* The y-axis in both figures represents S-IR, while the x-axes represent F-ER and CF-ER, respectively. We focus on S-IR because, in this puzzle, the cause is more sufficient than necessary for producing the effect.

**Results.** Figure 6 shows the sufficiency inconsistency rate (S-IR) in relation to the factual/counterfactual error rates (F/CF-ER) across all approaches.[6] SFT and DPO models, trained exclusively on either factual or counterfactual examples (*SFT+OnlyF*, *SFT+OnlyCF*, *DPO+OnlyF*, and *DPO+OnlyCF*) do not improve S-IR, even though they manage to reduce the corresponding F/CF-ER. However, when given access to both types of examples, *DPO+F&CF* shows an improvement in S-IR, though this improvement is not as pronounced as the reduction observed in F/CF-ER, particularly in CF-ER. The *SFT+F&CF* model shows a significant enhancement in both S-IR and F-ER, but it fails to make progress in CF-ER. Finally, by directly addressing causal consistency, with S-IR factored into the reward during fine-tuning, the *DPO+CCF* model achieves substantial improvements across F-ER, CF-ER, and S-IR. These results highlight the crucial role of effectively coordinating factual and counterfactual feedback for advanced reasoning tasks.

## 5.2 MODES OF GENERALIZATION

In this section, we answer the question "*to what extent the performance improvements in causal reasoning achieved through the fine-tuning process generalize across all the generalization modes defined in Section 2*". As mentioned in Section 2, an in-domain evaluation alone is inadequate for fully assessing the success of fine-tuning for reasoning and differentiating it from basic recall. Therefore, we evaluate all fine-tuning methods in the generalization modes introduced in Section 2.

**Experimental Setup.** To allow for the problem in Figure 5 to reflect all possible generalization modes we made slight modifications to the puzzle context, creating two variations: chain NDE and chain WDE (refer to Structure-2 and Structure-3 in Appendix D.1). The top section in Figure 7

---

[6] When evaluating models, we sample 10 answers for each question, which gives us a distribution over ER/IR (rather than just a point estimation).

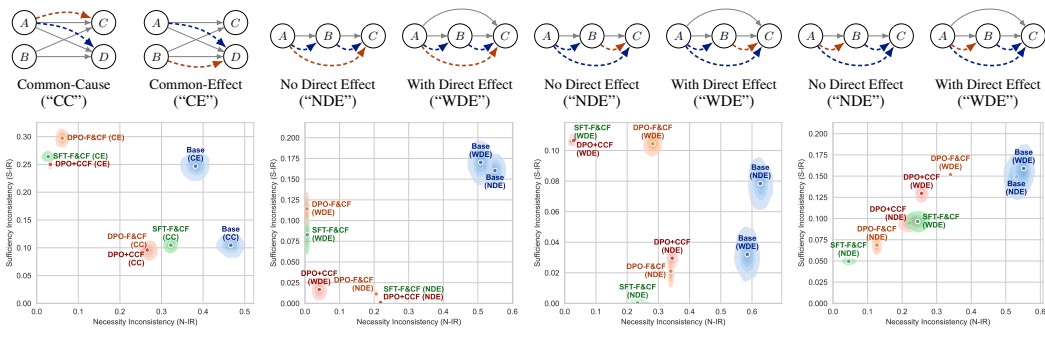

**(a) Common-Cause/Effect**    **(b) Inductive**    **(c) Deductive, Cause-B.**    **(d) Deductive, Effect-B.**

Figure 7: *Generalization results in the candy party puzzle.* **Top:** Eight scenarios involving three different causal structures: the bipartite graph $\{A, B\} \rightarrow \{C, D\}$ (Structure-1 in Appendix D.1) as well as the chain $A \rightarrow B \rightarrow C$ with and without a direct effect from $A$ to $C$ (Structure-2 and Structure-3 in Appendix D.1). **Blue** and **orange** arrows show the cause-effect interventions demonstrated to the model during fine-tuning and evaluation phases. **Bottom:** The causal reasoning ability of the fine-tuned models generalizes most effectively in inductive demonstrations. However, with common-cause/effect and deductive demonstrations, they no longer show the same reasoning improvements as observed in the in-domain setting.

displays all the causal structures used for each generalization mode, along with the cause-effect interventions demonstrated during the fine-tuning and evaluation phases. Based on the findings from the in-domain reasoning experiments (Section 5.1), where both SFT and DPO fine-tuning methods showed significantly better performance when provided with both factual and counterfactual examples, we include here only the methods *SFT-F&CF*, *DPO-F&CF*, *DPO+CCF*, and the ***Base*** model.

**Results.** The bottom section in Figure 7 presents the causal reasoning performance of all systems across the different generalization modes. We observe that: **(i) Common-Cause/Effect.** Fine-tuning based on demonstrations that involve just the target cause or the target effect (but not both as in the in-domain case) no longer leads to improvements in S-IR (unlike the in-domain case). While we do see improvements in N-IR, this can be attributed to better recall and not necessarily to better reasoning. The common-effect case leads to the greater improvement in N-IR precisely because the task of identifying factuals remains the same in this mode of generalization. **(ii) Induction.** Fine-tuning generalizes best when performed inductively. This is because relationships involving both the target cause and the target effect have been demonstrated, albeit not together. **(iii) Deductions.** While harder than induction, deduction is also possible as long as there are no direct effects that circumvent the intermediate variable. If there are such effects, deduction based on a shared cause becomes virtually impossible: Without any intervention on the intermediate variable, it is challenging to tell how much of the shared cause's effect is mediated through the intermediate variable vs. how much of it is not. Meanwhile, this seems to be identifiable to some extent when interventions on the intermediate variable are demonstrated as in deduction based on a shared effect (see Appendix H for a more detailed analysis).

## 5.3 REAL-WORLD PROBLEMS

**Experimental Setup.** We present three real-world causal reasoning problems: in the **Healthcare** domain, we examine breast cancer treatment and develop a simplified problem that determines how different treatment options—namely, radiotherapy/chemotherapy and surgery—are assigned to patients based on cancer type, tumor size, and nodal involvement. This model is grounded in a real-world guideline (MD Anderson Cancer Center) and published statistics on the disease (Orrantia-Borunda et al., 2022; Sezgin et al., 2020; Carey et al., 2006). In the **Engineering** domain, we implement an automatic fault detection algorithm for transmission lines (Reddy et al., 2016). This algorithm aims to identify the type of fault occurring on a transmission line using three different measurements. In the **Math Benchmarking** domain, we select a math question from GSM8K (Cobbe et al., 2021), a widely used benchmark for evaluating language models on grade school math problems. A detailed explanation of these three problems, including the context, factual and counterfactual questions, causal structures, and the cause-effect interventions demonstrated during the fine-tuning and evaluation phases across different generalization modes, can be found in Appendix D.2, D.3, D.4 respectively. For the real-world problems, we sample the same number of contexts for each method as this is more likely to be the case in a real-world application, where each context would correspond to an individual entry (e.g. a single patient in the healthcare domain, see Appendix B for details).

**Results.** The results for all three problems across in-domain and different generalization modes are available in Table 2 in Appendix A. Given the extensive number of experiments in this table, we have summarized the *Average Error Rate* (Avg-ER) and *Average Inconsistency Rate* (Avg-IR) scores in Table 1. For this summary, we first normalized the scores of each approach relative to the scores of the corresponding *Base* approach. Then, for each generalization mode (including the in-domain scenario), we calculated the average score of each tested method across all applicable problems. Not all generalization modes can be tested for every problem due to differences in causal structures, and the average score includes only the problems that were tested for each generalization

Table 1: *Average generalization performance* across three real-world causal reasoning problems. The scores are normalized relative to the *Base* approach's scores in each generalization mode. Higher scores indicate a greater number of errors made by the approach, with scores above 1.0 meaning that the approach makes more mistakes than the *Base* model, which has not undergone any fine-tuning.

| Mode | Metric | Base | OnlyF | | F&CF (Ours) | | |
|---|---|---|---|---|---|---|---|
| | | Base | SFT | DPO | SFT | DPO | DPO+CCF |
| In-Domain | Avg-ER | 1.00 | 0.82 | 0.86 | 0.42 | 0.53 | 0.48 |
| | Avg-IR | 1.00 | 0.73 | 0.72 | 0.44 | 0.51 | 0.47 |
| Common-Cause | Avg-ER | 1.00 | 1.35 | 1.75 | 1.17 | 1.62 | 2.04 |
| | Avg-IR | 1.00 | 1.30 | 1.49 | 1.14 | 1.42 | 1.80 |
| Common-Effect | Avg-ER | 1.00 | 0.60 | 0.71 | 0.53 | 0.66 | 0.64 |
| | Avg-IR | 1.00 | 0.49 | 0.60 | 0.42 | 0.53 | 0.50 |
| Inductive | Avg-ER | 1.00 | 0.86 | 0.79 | 0.60 | 0.58 | 0.69 |
| | Avg-IR | 1.00 | 0.74 | 0.62 | 0.51 | 0.50 | 0.59 |
| Deductive, Cause-B. | Avg-ER | 1.00 | 0.70 | 0.67 | 0.61 | 0.58 | 0.57 |
| | Avg-IR | 1.00 | 0.62 | 0.59 | 0.53 | 0.53 | 0.51 |
| Deductive, Effect-B. | Avg-ER | 1.00 | 0.83 | 0.83 | 0.90 | 0.95 | 0.77 |
| | Avg-IR | 1.00 | 0.91 | 0.87 | 0.93 | 0.89 | 0.76 |

mode. In Table 1, higher scores indicate more errors, and scores above 1.0 signify that the approach makes more mistakes than the *Base* model. We observe that: (i) In the in-domain scenario, when the fine-tuning is guided by both factual and counterfactual examples (-*F&CF*), the language models show a significant improvement in causal reasoning ability. (ii) Similar to what we observed in previous experiments, this improvement generalizes to most generalization modes, with the exception of *common-cause* and *effect-based deduction*. (iii) In most of modes, language models trained with causal consistency feedback (*DPO+CCF*) demonstrate a lower error and inconsistency rate.

## 6 RELATED WORK

**Reasoning Evaluation.** While our work focuses on reasoning *elicitation*, there is a plethora of work on reasoning *evaluation* (Frohberg and Binder, 2021; Wu et al., 2023; Chang et al., 2024). Parmar et al. (2024) evaluate logical reasoning, Cohn and Hernandez-Orallo (2023) evaluate spatial reasoning, Gandhi et al. (2024) evaluate social reasoning, Li et al. (2022); Jin et al. (2023); Ashwani et al. (2024); Li et al. (2024); Wang (2024) evaluate causal reasoning, and Maasch et al. (2025) evaluate compositional reasoning. Being able to determine which models are better at reasoning is an important aspect of reasoning elicitation. We have explored this aspect in Section 3 building on the work of González and Nori (2024), which has allowed us to consider relationships like necessity and sufficiency that require a higher level of reasoning than simply answering counterfactual prompts.

**Counterfactual Frameworks.** Counterfactual frameworks have been employed to explore aspects of LLMs other than reasoning elicitation. For instance, Lin et al. (2024b) formulate preference alignment as a causal inference problem and develop an alternative approach to algorithms like DPO. In Wu et al. (2021); Nguyen et al. (2024), counterfactual inputs are used to explain a model's predictions. Additionally, Kandpal et al. (2023); Zhang et al. (2023b) model memorization through counterfactuals.

**Fine-tuning with Factual Feedback.** While we fine-tune language models for reasoning with counterfactual feedback, previous work has considered fine-tuning for factuality—that is providing factually correct answers to questions which do not involve interventions (Tian et al., 2023; Tong et al., 2024; Lin et al., 2024a). Khalifa et al. (2020); Korbak et al. (2022a;b) propose methods for *controlled generation*, which aim to constrain a model's answers using binary reward functions. While they suggest using correctness as the reward function to improve factuality, we have shown that targeting metrics like causal consistency require more fine-grained feedback (beyond a binary reward).

## 7 CONCLUSION

This work introduced the problem of *fine-tuning for reasoning*, along with (i) a taxonomy for generalization modes, (ii) multiple metrics that address the limitations of existing performance measures, and (iii) methods for generating fine-tuning data with counterfactual feedback. We showed that fine-tuning for reasoning requires both factual and counterfactual examples, and that paring examples related through a shared context can lead to improvements. A key limitation of our approach is the restriction of causes and effects to binary variables, which allows us to focus on high-level relationships like necessity and sufficiency present in human reasoning (see Appendix I for further discussion).

ACKNOWLEDGMENTS

Author J. Maasch acknowledges the National Science Foundation Graduate Research Fellowship under Grant No. DGE–2139899.

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

# A  BREAKDOWN OF THE RESULTS IN SECTION 5.3

Table 2: *Results of Healthcare, Engineering, and Math Benchmarking problems.* For some scenarios in *Math Benchmarking*, N-IR and AS-IR are equal to 0.00 for all algorithms because the target cause $X$ is never present without an intervention due to how these scenarios are structured.

| Scenario | | Alg. | Correctness | | | Causal Consistency | | | | |
|---|---|---|---|---|---|---|---|---|---|---|
| | | | F-ER | C-ER | Avg-ER | N-IR | S-IR | AN-IR | AS-IR | Avg-IR |
| **Healthcare** Breast Cancer Treatment | | Base | 23.57(0.00) | 28.93(0.00) | 26.25(0.00) | 20.62(0.00) | 4.34(0.00) | 11.47(0.00) | 32.01(0.00) | 17.11(0.00) |
| | In-domain | SFT-OnlyF | 3.12(0.04) | 20.82(0.02) | 11.97(0.02) | 1.50(0.01) | 2.09(0.02) | 3.01(0.02) | 19.93(0.01) | 6.63(0.01) |
| | | DPO-OnlyF | 10.77(1.83) | 28.57(2.07) | 19.67(1.94) | 9.10(2.24) | 4.76(0.10) | 6.00(0.16) | 25.32(1.10) | 11.29(0.66) |
| | | SFT-F&CF | 1.64(0.01) | 0.08(0.00) | 0.86(0.00) | 0.85(0.00) | 0.80(0.00) | 0.82(0.00) | 0.87(0.00) | 0.84(0.00) |
| | | DPO-F&CF | 12.53(5.04) | 8.74(1.69) | 10.63(3.14) | 8.65(3.04) | 4.32(0.30) | 5.00(0.28) | 11.07(4.93) | 7.26(1.58) |
| | | DPO+CCF | 9.55(1.16) | 5.16(0.15) | 7.36(0.52) | 7.97(1.21) | 1.70(0.01) | 3.59(0.04) | 9.31(1.51) | 5.64(0.36) |
| | Com-Cause | SFT-OnlyF | 48.86(3.27) | 38.82(0.91) | 43.84(1.65) | 42.38(2.82) | 10.55(0.10) | 13.96(0.03) | 45.71(2.06) | 28.15(0.65) |
| | | DPO-OnlyF | 64.56(1.93) | 67.95(0.19) | 66.25(0.70) | 53.01(1.45) | 13.20(0.06) | 14.25(0.00) | 64.77(0.58) | 36.31(0.28) |
| | | SFT-F&CF | 25.99(0.55) | 38.14(0.81) | 32.07(0.07) | 25.83(0.35) | 4.10(0.04) | 13.07(0.01) | 41.42(0.38) | 21.10(0.06) |
| | | DPO-F&CF | 53.92(2.45) | 63.62(4.52) | 58.77(2.10) | 46.27(0.44) | 14.09(0.01) | 14.44(0.04) | 53.23(2.34) | 32.01(0.33) |
| | | DPO+CCF | 80.18(4.78) | 72.85(0.21) | 76.52(1.55) | 70.55(1.67) | 15.96(0.03) | 14.93(0.00) | 68.67(3.67) | 42.53(0.69) |
| | Com-Effect | SFT-OnlyF | 3.16(0.04) | 20.87(0.02) | 12.01(0.02) | 1.45(0.01) | 2.21(0.02) | 2.94(0.03) | 20.06(0.01) | 6.67(0.01) |
| | | DPO-OnlyF | 5.93(0.02) | 30.30(2.44) | 18.11(0.54) | 11.75(2.24) | 3.77(0.04) | 4.61(0.00) | 21.60(0.03) | 10.43(0.12) |
| | | SFT-F&CF | 1.45(0.01) | 20.43(0.02) | 10.94(0.00) | 1.60(0.00) | 0.53(0.00) | 1.40(0.00) | 19.68(0.01) | 5.80(0.00) |
| | | DPO-F&CF | 2.96(0.03) | 22.44(0.02) | 12.70(0.00) | 1.40(0.01) | 2.08(0.03) | 4.14(0.01) | 19.84(0.00) | 6.86(0.00) |
| | | DPO+CCF | 2.17(0.04) | 22.78(0.11) | 12.48(0.07) | 0.36(0.00) | 2.58(0.07) | 3.16(0.05) | 19.60(0.00) | 6.42(0.02) |
| | Deductive (Cause-Based) | SFT-OnlyF | 1.56(0.01) | 20.30(0.01) | 10.93(0.01) | 1.14(0.00) | 0.65(0.01) | 1.40(0.01) | 20.12(0.00) | 5.83(0.00) |
| | | DPO-OnlyF | 18.52(4.41) | 32.93(4.36) | 25.73(4.34) | 13.03(2.76) | 6.52(0.21) | 6.80(0.20) | 30.30(2.82) | 14.16(1.11) |
| | | SFT-F&CF | 1.32(0.00) | 22.37(0.00) | 11.84(0.00) | 1.28(0.00) | 0.15(0.00) | 3.07(0.00) | 20.48(0.00) | 6.24(0.00) |
| | | DPO-F&CF | 4.25(0.14) | 24.37(0.14) | 14.31(0.12) | 1.45(0.04) | 3.21(0.09) | 5.99(0.09) | 20.36(0.03) | 7.75(0.04) |
| | | DPO+CCF | 5.05(0.04) | 25.17(0.17) | 15.11(0.07) | 2.95(0.03) | 3.97(0.07) | 6.42(0.08) | 20.12(0.00) | 8.36(0.02) |
| **Engineering** Transmission Line Protection | | Base | 16.39(0.00) | 27.10(0.00) | 21.75(0.00) | 13.70(0.00) | 27.52(0.00) | 2.71(0.00) | 13.96(0.00) | 14.47(0.00) |
| | In-domain | SFT-OnlyF | 2.94(0.05) | 29.64(0.06) | 16.29(0.01) | 4.84(0.02) | 26.73(0.00) | 0.94(0.02) | 2.18(0.01) | 8.67(0.00) |
| | | DPO-OnlyF | 3.76(0.05) | 31.30(0.06) | 17.53(0.00) | 6.63(0.01) | 26.42(0.00) | 1.00(0.00) | 3.23(0.05) | 9.32(0.00) |
| | | SFT-F&CF | 1.39(0.00) | 6.43(0.05) | 3.91(0.01) | 3.27(0.04) | 4.32(0.13) | 0.46(0.00) | 0.93(0.00) | 2.25(0.00) |
| | | DPO-F&CF | 10.23(2.60) | 33.48(1.15) | 21.86(1.76) | 6.94(0.29) | 30.13(0.77) | 5.34(1.23) | 6.87(0.82) | 12.32(0.72) |
| | | DPO+CCF | 7.64(0.89) | 23.00(0.60) | 15.32(0.71) | 6.61(0.12) | 19.61(0.77) | 4.44(0.32) | 4.03(0.37) | 8.67(0.32) |
| | Com-Cause | SFT-OnlyF | 14.61(0.08) | 30.28(0.17) | 22.44(0.11) | 13.80(0.08) | 27.23(0.00) | 0.81(0.00) | 14.02(0.07) | 13.96(0.02) |
| | | DPO-OnlyF | 11.25(0.15) | 30.78(0.20) | 21.02(0.18) | 11.04(0.15) | 26.60(0.00) | 0.21(0.00) | 11.73(0.16) | 12.39(0.04) |
| | | SFT-F&CF | 14.05(0.10) | 34.43(0.15) | 24.24(0.12) | 13.13(0.06) | 32.64(0.18) | 0.92(0.01) | 13.60(0.06) | 15.07(0.05) |
| | | DPO-F&CF | 12.52(0.17) | 31.46(0.10) | 21.99(0.11) | 10.84(0.12) | 30.40(0.12) | 1.72(0.08) | 11.41(0.11) | 13.84(0.06) |
| | | DPO+CCF | 14.20(0.09) | 35.72(0.00) | 24.96(0.02) | 13.26(0.07) | 35.68(0.01) | 0.94(0.01) | 13.56(0.06) | 15.86(0.02) |
| | Com-Effect | SFT-OnlyF | 2.71(0.05) | 29.26(0.04) | 15.99(0.01) | 4.39(0.03) | 26.73(0.00) | 0.78(0.01) | 2.07(0.02) | 8.49(0.01) |
| | | DPO-OnlyF | 2.71(0.05) | 29.26(0.04) | 15.99(0.01) | 4.39(0.03) | 26.73(0.00) | 0.78(0.01) | 2.07(0.02) | 8.49(0.01) |
| | | SFT-F&CF | 0.95(0.00) | 26.41(0.00) | 13.68(0.00) | 0.82(0.00) | 26.41(0.00) | 0.15(0.00) | 0.80(0.00) | 7.04(0.00) |
| | | DPO-F&CF | 2.28(0.09) | 34.30(1.84) | 18.29(0.48) | 3.04(0.10) | 25.71(0.02) | 0.46(0.00) | 8.74(1.32) | 9.49(0.13) |
| | | DPO+CCF | 1.51(0.00) | 32.98(2.12) | 17.25(0.49) | 2.41(0.04) | 26.55(0.00) | 0.22(0.00) | 6.50(1.23) | 8.92(0.11) |
| | Inductive | SFT-OnlyF | 2.58(0.02) | 31.10(0.10) | 16.84(0.03) | 5.85(0.07) | 26.41(0.00) | 0.57(0.00) | 2.38(0.03) | 8.80(0.01) |
| | | DPO-OnlyF | 2.68(0.05) | 31.36(0.07) | 17.02(0.00) | 6.22(0.00) | 25.69(0.01) | 0.46(0.00) | 2.89(0.06) | 8.81(0.00) |
| | | SFT-F&CF | 1.43(0.00) | 20.25(0.17) | 10.84(0.04) | 0.91(0.00) | 20.53(0.16) | 0.52(0.00) | 0.91(0.00) | 5.72(0.01) |
| | | DPO-F&CF | 2.85(0.02) | 22.23(0.69) | 12.54(0.22) | 2.53(0.03) | 22.27(0.67) | 0.33(0.00) | 2.56(0.03) | 6.92(0.07) |
| | | DPO+CCF | 2.34(0.06) | 26.23(0.01) | 14.29(0.02) | 1.37(0.01) | 26.03(0.01) | 1.05(0.04) | 1.47(0.01) | 7.48(0.01) |
| **Math Benchmarking** Testing on: $S \rightarrow T$ | | Base | 47.35(0.00) | 26.32(0.00) | 36.84(0.00) | 0.00(0.00) | 61.50(0.00) | 48.24(0.00) | 0.00(0.00) | 27.44(0.00) |
| | In-domain | SFT-OnlyF | 37.21(0.58) | 28.91(0.08) | 33.06(0.25) | 0.00(0.00) | 52.17(0.41) | 40.10(0.69) | 0.00(0.00) | 23.07(0.14) |
| | | DPO-OnlyF | 28.28(0.95) | 57.39(3.22) | 42.83(1.84) | 0.00(0.00) | 66.84(1.15) | 29.78(0.79) | 0.00(0.00) | 24.15(0.24) |
| | | SFT-F&CF | 22.52(0.18) | 14.05(0.17) | 18.29(0.07) | 0.00(0.00) | 32.07(0.22) | 23.45(0.18) | 0.00(0.00) | 13.88(0.05) |
| | | DPO-F&CF | 14.06(0.10) | 26.87(1.71) | 20.47(0.32) | 0.00(0.00) | 37.78(1.52) | 15.90(0.07) | 0.00(0.00) | 13.42(0.07) |
| | | DPO+CCF | 19.38(0.04) | 36.13(0.44) | 27.75(0.19) | 0.00(0.00) | 50.89(1.13) | 21.03(0.02) | 0.00(0.00) | 17.98(0.09) |
| | Inductive | SFT-OnlyF | 37.25(0.78) | 33.46(0.02) | 35.36(0.17) | 0.00(0.00) | 53.39(0.46) | 42.11(0.62) | 0.00(0.00) | 23.88(0.13) |
| | | DPO-OnlyF | 15.31(0.27) | 42.93(0.01) | 29.12(0.10) | 0.00(0.00) | 52.54(0.31) | 18.06(0.28) | 0.00(0.00) | 17.65(0.07) |
| | | SFT-F&CF | 24.27(0.08) | 27.13(0.20) | 25.70(0.04) | 0.00(0.00) | 40.98(0.05) | 27.00(0.11) | 0.00(0.00) | 16.99(0.02) |
| | | DPO-F&CF | 16.13(1.16) | 26.86(1.39) | 21.49(0.31) | 0.00(0.00) | 38.28(0.95) | 18.44(1.08) | 0.00(0.00) | 14.18(0.18) |
| | | DPO+CCF | 22.21(1.33) | 29.94(1.19) | 26.07(0.34) | 0.00(0.00) | 48.44(1.23) | 23.51(1.16) | 0.00(0.00) | 17.99(0.23) |
| **Math Benchmarking** Testing on: $S \rightarrow R$ | | Base | 50.25(0.00) | 51.11(0.00) | 50.68(0.00) | 0.00(0.00) | 67.30(0.00) | 58.65(0.00) | 0.00(0.00) | 31.49(0.00) |
| | In-domain | SFT-OnlyF | 45.16(0.01) | 56.90(0.13) | 51.03(0.03) | 0.00(0.00) | 68.02(0.02) | 52.95(0.05) | 0.00(0.00) | 30.24(0.00) |
| | | DPO-OnlyF | 39.11(0.06) | 72.29(0.45) | 55.70(0.16) | 0.00(0.00) | 76.30(0.06) | 40.57(0.06) | 0.00(0.00) | 29.22(0.00) |
| | | SFT-F&CF | 44.55(0.01) | 32.88(0.09) | 38.71(0.01) | 0.00(0.00) | 62.18(0.00) | 47.15(0.01) | 0.00(0.00) | 27.33(0.00) |
| | | DPO-F&CF | 15.43(3.99) | 8.56(0.68) | 11.99(1.99) | 0.00(0.00) | 20.06(5.01) | 16.48(3.99) | 0.00(0.00) | 9.14(1.12) |
| | | DPO+CCF | 17.12(1.78) | 8.06(0.87) | 12.59(1.26) | 0.00(0.00) | 23.48(4.74) | 17.88(1.72) | 0.00(0.00) | 10.34(0.75) |
| | Deductive (Effect-Based) | SFT-OnlyF | 46.17(0.03) | 42.46(0.18) | 44.31(0.05) | 0.00(0.00) | 66.31(0.04) | 49.46(0.10) | 0.00(0.00) | 28.94(0.00) |
| | | DPO-OnlyF | 39.36(0.78) | 48.96(2.24) | 44.16(0.38) | 0.00(0.00) | 65.83(0.41) | 44.21(1.15) | 0.00(0.00) | 27.51(0.06) |
| | | SFT-F&CF | 46.57(0.04) | 48.58(1.08) | 47.57(0.22) | 0.00(0.00) | 66.05(0.10) | 52.63(0.05) | 0.00(0.00) | 29.67(0.01) |
| | | DPO-F&CF | 39.50(0.00) | 61.68(4.62) | 50.59(1.15) | 0.00(0.00) | 73.43(0.47) | 39.61(0.00) | 0.00(0.00) | 28.26(0.03) |
| | | DPO+CCF | 31.05(0.96) | 50.27(4.58) | 40.66(1.93) | 0.00(0.00) | 63.75(2.23) | 32.92(0.80) | 0.00(0.00) | 24.17(0.34) |
| **Math Benchmarking** Testing on: $R \rightarrow T$ | | Base | 47.35(0.00) | 58.82(0.00) | 53.08(0.00) | 28.77(0.00) | 50.11(0.00) | 25.69(0.00) | 22.44(0.00) | 31.76(0.00) |
| | In-domain | SFT-OnlyF | 36.07(0.97) | 65.96(0.02) | 51.01(0.30) | 21.32(0.09) | 54.11(0.02) | 26.60(0.27) | 9.97(0.25) | 28.00(0.13) |
| | | DPO-OnlyF | 28.28(0.95) | 22.81(0.12) | 25.54(0.42) | 32.30(0.31) | 2.88(0.01) | 0.24(0.00) | 28.46(0.92) | 15.97(0.14) |
| | | SFT-F&CF | 23.12(0.12) | 45.03(0.13) | 34.08(0.11) | 16.87(0.02) | 39.39(0.09) | 18.67(0.03) | 4.76(0.04) | 19.92(0.04) |
| | | DPO-F&CF | 24.33(2.42) | 22.38(0.16) | 23.35(0.47) | 30.27(0.30) | 6.11(0.45) | 3.74(0.46) | 21.18(1.63) | 15.32(0.39) |
| | | DPO+CCF | 21.03(0.05) | 20.91(0.55) | 20.97(0.16) | 27.75(0.11) | 5.52(0.17) | 0.83(0.01) | 20.58(0.06) | 13.67(0.03) |
| | Deductive (Cause-Based) | SFT-OnlyF | 37.87(0.81) | 62.75(0.10) | 50.31(0.36) | 23.38(0.11) | 51.63(0.07) | 26.27(0.28) | 12.13(0.25) | 28.35(0.14) |
| | | DPO-OnlyF | 15.31(0.27) | 19.98(0.03) | 17.65(0.08) | 23.77(0.23) | 4.04(0.02) | 0.24(0.00) | 15.61(0.28) | 10.91(0.06) |
| | | SFT-F&CF | 25.41(0.10) | 53.63(0.15) | 39.52(0.05) | 19.12(0.04) | 44.40(0.03) | 18.22(0.07) | 7.55(0.12) | 22.32(0.02) |
| | | DPO-F&CF | 24.33(1.01) | 38.62(6.13) | 31.48(2.94) | 29.16(0.35) | 22.64(5.93) | 0.66(0.00) | 24.69(1.07) | 19.29(1.00) |
| | | DPO+CCF | 15.97(0.42) | 40.45(3.76) | 28.21(1.45) | 23.14(0.23) | 25.35(3.48) | 1.85(0.02) | 14.88(0.56) | 16.31(0.46) |

# B NUMBER OF CONTEXT SAMPLES VS. QUESTION-ANSWER EXAMPLES

When collecting datasets, we always sample the same number of context variables $U$ across all feedback types, whether it be *OnlyF*, *OnlyCF*, *F&CF* or *CCF*. This is with the exception of Section 5.1 where we double the number of contexts for OnlyF and OnlyCF to end up with the same number question-answer examples as other methods. In other words, for those experiments, F&C has access to $N$ context samples with two question-answer examples for each context sample (one factual, one counterfactual). Meanwhile, OnlyF and OnlyCF have access to $2N$ context samples with a single question-answer example for each context (either factual or counterfactual).

This means, for the remainder of the experiments, F&CF has more question-answer examples in its training dataset, but crucially, each method still sees the same number of problem instances. For example, in the healthcare domain, OnlyF and F&CF sees the same set of breast cancer patients (each patient corresponding to a context); the training data for F&CF does not include any new patients; and the additional CF feedback is provided only for the existing patients. Therefore, the comparison between OnlyF and F&CF remains fair, which would not have been the case if the context variables for the factual dataset and the counterfactual dataset were sampled independently.

In Section 5.1, we preferred the setup with the same number of question-answer examples across methods to be absolutely certain that any improvement we see is due to the type of feedback being provided. In Sections 5.2 and 5.3, we preferred the setup with the same number of context variables across methods because this is likely to be the case in practice (for instance, if we apply these methods in a healthcare setting, the number of patients would be constant for each method).

That being said, for completeness, we report here supplementary generalization results where OnlyF has access to twice the number of context variables, hence the same number of question-answer examples (as in Section 5.1). We call these new baselines *OnlyFx2* to differentiate them from the original setup, see Table 3 for a detailed breakdown and Table 4 for a summary.

Table 3: *Detailed breakdown of the generalization results for SFT-OnlyFx2 and DPO-OnlyFx2.*

| Scenario | | Alg. | Correctness | | | Causal Consistency | | | | |
|---|---|---|---|---|---|---|---|---|---|---|
| | | | F-ER | C-ER | Avg-ER | N-IR | S-IR | AN-IR | AS-IR | Avg-IR |
| **Healthcare** Breast Cancer Treatment | | Base | 23.57(0.00) | 28.93(0.00) | 26.25(0.00) | 20.62(0.00) | 4.34(0.00) | 11.47(0.00) | 32.01(0.00) | 17.11(0.00) |
| | In-domain | SFT-OnlyFx2 | 3.09(0.01) | 21.41(0.01) | 12.25(0.01) | 2.39(0.01) | 1.15(0.00) | 2.82(0.01) | 21.06(0.01) | 6.86(0.01) |
| | | DPO-OnlyFx2 | 5.18(0.32) | 24.70(0.61) | 14.94(0.45) | 4.21(0.40) | 2.75(0.05) | 5.09(0.28) | 21.68(0.13) | 8.43(0.19) |
| | Com-Cause | SFT-OnlyFx2 | 31.45(1.57) | 26.48(0.66) | 28.97(1.01) | 24.84(0.79) | 7.86(0.21) | 9.75(0.21) | 34.45(0.42) | 19.22(0.35) |
| | | DPO-OnlyFx2 | 76.27(4.04) | 68.74(0.09) | 72.51(0.99) | 65.14(3.45) | 13.27(0.06) | 14.69(0.00) | 70.43(0.86) | 40.88(0.55) |
| | Com-Effect | SFT-OnlyFx2 | 2.58(0.01) | 21.05(0.03) | 11.82(0.01) | 2.01(0.01) | 1.01(0.00) | 2.58(0.02) | 20.54(0.00) | 6.54(0.01) |
| | | DPO-OnlyFx2 | 2.23(0.02) | 22.43(0.03) | 12.33(0.03) | 1.23(0.01) | 1.80(0.02) | 3.44(0.03) | 19.76(0.00) | 6.56(0.01) |
| | Deductive (Cause-Based) | SFT-OnlyFx2 | 2.26(0.00) | 22.26(0.02) | 12.26(0.01) | 1.95(0.01) | 0.63(0.00) | 3.10(0.02) | 20.91(0.00) | 6.65(0.00) |
| | | DPO-OnlyFx2 | 3.57(0.03) | 22.99(0.02) | 13.28(0.00) | 1.66(0.01) | 2.67(0.02) | 4.44(0.01) | 20.30(0.02) | 7.27(0.00) |
| **Engineering** Tx. Line Protection | | Base | 16.39(0.00) | 27.10(0.00) | 21.75(0.00) | 13.70(0.00) | 27.52(0.00) | 2.71(0.00) | 13.96(0.00) | 14.47(0.00) |
| | In-domain | SFT-OnlyFx2 | 0.44(0.00) | 31.65(0.02) | 16.05(0.00) | 5.59(0.01) | 26.40(0.00) | 0.07(0.00) | 0.40(0.00) | 8.11(0.00) |
| | | DPO-OnlyFx2 | 4.21(0.06) | 30.13(0.00) | 17.17(0.01) | 5.54(0.00) | 26.77(0.01) | 0.92(0.03) | 3.49(0.02) | 9.18(0.01) |
| | Com-Cause | SFT-OnlyFx2 | 16.35(0.00) | 35.69(0.00) | 26.02(0.00) | 15.88(0.00) | 26.86(0.00) | 0.47(0.00) | 16.05(0.01) | 14.82(0.00) |
| | | DPO-OnlyFx2 | 13.52(0.02) | 33.53(0.01) | 23.53(0.00) | 13.29(0.03) | 26.65(0.00) | 0.24(0.00) | 15.13(0.00) | 13.83(0.00) |
| | Com-Effect | SFT-OnlyFx2 | 0.43(0.00) | 32.38(0.00) | 16.40(0.00) | 6.21(0.00) | 26.40(0.00) | 0.07(0.00) | 0.44(0.00) | 8.28(0.00) |
| | | DPO-OnlyFx2 | 3.01(0.09) | 29.61(0.09) | 16.31(0.03) | 5.13(0.08) | 26.69(0.00) | 0.44(0.00) | 2.69(0.09) | 8.74(0.02) |
| | Inductive | SFT-OnlyFx2 | 0.94(0.00) | 32.55(0.00) | 16.75(0.00) | 6.51(0.00) | 26.32(0.00) | 0.43(0.00) | 0.75(0.00) | 8.50(0.00) |
| | | DPO-OnlyFx2 | 1.72(0.01) | 31.98(0.00) | 16.85(0.00) | 6.11(0.00) | 26.37(0.00) | 0.08(0.00) | 1.68(0.01) | 8.56(0.00) |
| **Math Bench.** $S \to T$ | | Base | 47.35(0.00) | 26.32(0.00) | 36.84(0.00) | 0.00(0.00) | 61.50(0.00) | 48.24(0.00) | 0.00(0.00) | 27.44(0.00) |
| | In-domain | SFT-OnlyFx2 | 26.59(0.38) | 26.57(0.01) | 26.58(0.13) | 0.00(0.00) | 42.88(0.31) | 28.33(0.47) | 0.00(0.00) | 17.80(0.10) |
| | | DPO-OnlyFx2 | 24.47(0.41) | 48.01(1.30) | 36.24(0.61) | 0.00(0.00) | 59.29(0.82) | 28.08(0.39) | 0.00(0.00) | 21.84(0.14) |
| | Inductive | SFT-OnlyFx2 | 22.30(0.08) | 28.04(0.03) | 25.17(0.03) | 0.00(0.00) | 40.63(0.06) | 23.93(0.09) | 0.00(0.00) | 16.14(0.02) |
| | | DPO-OnlyFx2 | 16.72(1.55) | 46.65(2.01) | 31.69(1.70) | 0.00(0.00) | 52.99(1.61) | 18.49(1.35) | 0.00(0.00) | 17.87(0.37) |
| **Math Bench.** $S \to R$ | | Base | 50.25(0.00) | 51.11(0.00) | 50.68(0.00) | 0.00(0.00) | 67.30(0.00) | 58.65(0.00) | 0.00(0.00) | 31.49(0.00) |
| | In-domain | SFT-OnlyFx2 | 35.95(0.36) | 56.36(0.08) | 46.15(0.16) | 0.00(0.00) | 61.39(0.16) | 45.74(0.34) | 0.00(0.00) | 26.78(0.06) |
| | | DPO-OnlyFx2 | 39.06(0.84) | 60.64(1.60) | 49.85(0.56) | 0.00(0.00) | 69.24(0.39) | 44.19(1.08) | 0.00(0.00) | 28.36(0.09) |
| | Deductive (Effect-Based) | SFT-OnlyFx2 | 45.97(0.01) | 45.91(0.21) | 45.94(0.04) | 0.00(0.00) | 66.49(0.05) | 50.64(0.07) | 0.00(0.00) | 29.28(0.00) |
| | | DPO-OnlyFx2 | 24.75(2.64) | 52.68(2.02) | 38.72(1.88) | 0.00(0.00) | 61.52(1.88) | 27.44(2.60) | 0.00(0.00) | 22.24(0.54) |
| **Math Bench.** $R \to T$ | | Base | 47.35(0.00) | 58.82(0.00) | 53.08(0.00) | 28.77(0.00) | 50.11(0.00) | 25.69(0.00) | 22.44(0.00) | 31.76(0.00) |
| | In-domain | SFT-OnlyFx2 | 7.80(1.14) | 64.03(0.05) | 45.92(0.38) | 19.43(0.17) | 51.94(0.03) | 20.63(0.24) | 7.58(0.36) | 24.90(0.17) |
| | | DPO-OnlyFx2 | 24.47(0.41) | 24.54(0.18) | 24.50(0.08) | 29.80(0.21) | 7.40(0.44) | 1.74(0.06) | 23.25(0.64) | 15.55(0.04) |
| | Deductive (Cause-Based) | SFT-OnlyFx2 | 22.84(0.09) | 59.90(0.07) | 41.37(0.07) | 17.28(0.01) | 49.03(0.05) | 17.49(0.03) | 5.84(0.03) | 22.41(0.02) |
| | | DPO-OnlyFx2 | 16.72(1.55) | 22.86(0.27) | 19.79(0.70) | 24.01(0.63) | 6.87(0.16) | 1.15(0.01) | 15.90(1.59) | 11.98(0.27) |

Table 4: *Average generalization performance, including SFT-OnlyFx2 and DPO-OnlyFx2.*

| Mode | Metric | Base | OnlyF | | OnlyFx2 | | F&CF (Ours) | | |
|---|---|---|---|---|---|---|---|---|---|
| | | Base | SFT | DPO | SFT | DPO | SFT | DPO | DPO+CCF |
| In-Domain | Avg-ER | 1.00 | 0.82 | 0.86 | 0.74 | 0.76 | 0.42 | 0.53 | 0.48 |
| | Avg-IR | 1.00 | 0.73 | 0.72 | 0.65 | 0.66 | 0.44 | 0.51 | 0.47 |
| Common-Cause | Avg-ER | 1.00 | 1.35 | 1.75 | 1.14 | 1.92 | 1.17 | 1.62 | 2.04 |
| | Avg-IR | 1.00 | 1.30 | 1.49 | 1.07 | 1.67 | 1.14 | 1.42 | 1.80 |
| Common-Effect | Avg-ER | 1.00 | 0.60 | 0.71 | 0.60 | 0.60 | 0.53 | 0.66 | 0.64 |
| | Avg-IR | 1.00 | 0.49 | 0.60 | 0.47 | 0.49 | 0.42 | 0.53 | 0.50 |
| Inductive | Avg-ER | 1.00 | 0.86 | 0.79 | 0.72 | 0.81 | 0.60 | 0.58 | 0.69 |
| | Avg-IR | 1.00 | 0.74 | 0.62 | 0.58 | 0.62 | 0.51 | 0.50 | 0.59 |
| Deductive, Cause-B. | Avg-ER | 1.00 | 0.70 | 0.67 | 0.62 | 0.43 | 0.61 | 0.58 | 0.57 |
| | Avg-IR | 1.00 | 0.62 | 0.59 | 0.54 | 0.40 | 0.53 | 0.53 | 0.51 |
| Deductive, Effect-B. | Avg-ER | 1.00 | 0.83 | 0.83 | 0.95 | 0.78 | 0.90 | 0.95 | 0.77 |
| | Avg-IR | 1.00 | 0.91 | 0.87 | 0.82 | 0.73 | 0.93 | 0.89 | 0.76 |

# C  FORMAL DESCRIPTION OF THE ALGORITHMS IN SECTION 4

---
**Algorithm 1** Supervised Counterfactual Feedback
---
1: **Inputs:** Demonstrations $\mathcal{D} = \{\mathcal{P}_{X_i \to Y_i}\}$, question templates $q, \tilde{q}$, answer generator $H$
2: **Output:** Dataset $\mathbb{D} = \{q, a\}$ of questions and answer pairs

---
3: $\mathbb{D} \leftarrow \{\}$
4: **for** $\mathcal{P}_{X_i \to Y_i} \in \mathcal{D}, n \in \{1, \ldots, N\}$ **do**
5: $\quad u, X, Y, Y_{X'} \sim \mathcal{P}_{X_i \to Y_i}$ $\quad\quad\quad\quad\quad\quad$ ▷ Sample a context, cause, and potential effects
6: $\quad q_f \leftarrow q(u), q_{cf} \leftarrow \tilde{q}_{X'}(u)$ $\quad\quad\quad\quad\quad\quad\quad\quad$ ▷ Generate questions given the context
7: $\quad a_f \leftarrow H(q_f, Y_X), a_{cf} \leftarrow H(q_{cf}, Y_{X'})$ $\quad\quad$ ▷ Generate answers given the correct outcome
8: $\quad \mathbb{D} \leftarrow \mathbb{D} \cup \{(q_f, a_f), (q_{cf}, a_{cf})\}$
9: **end for**

---
**Algorithm 2** Preference-based Counterfactual Feedback
---
1: **Inputs:** Demonstrations $\mathcal{D} = \{\mathcal{P}_{X_i \to Y_i}\}$, question templates $q, \tilde{q}$, answer extractor $h$, target model $\ell_0$
2: **Output:** Dataset $\mathbb{D} = \{(q, a) \succ (q', a')\}$ of preferences over question-answer pairs

---
3: $\mathbb{D} \leftarrow \{\}$
4: **for** $\mathcal{P}_{X_i \to Y_i} \in \mathcal{D}, n \in \{1, \ldots, N\}$ **do**
5: $\quad u, X, Y, Y_{X'} \sim \mathcal{P}_{X_i \to Y_i}$ $\quad\quad\quad\quad\quad\quad$ ▷ Sample a context, cause, and potential effects
6: $\quad q_f \leftarrow q(u), q_{cf} \leftarrow \tilde{q}_{X'}(u)$ $\quad\quad\quad\quad\quad\quad\quad\quad$ ▷ Generate questions given the context
7: $\quad$ **for** $m \in \{1, \ldots, M\}$ **do**
8: $\quad\quad a_f[m] \sim \ell_0(q_f), a_{cf}[m] \sim \ell_0(q_{cf})$ $\quad\quad\quad$ ▷ Collect answers from the language model
9: $\quad\quad \hat{Y}_X[m] \leftarrow h(a_f[m]), \hat{Y}_{X'}[m] \leftarrow h(a_{cf}[m])$ $\quad$ ▷ Extract outcome estimates from answers
10: $\quad$ **end for**
11: $\quad$ **for** $m \in \{1, \ldots, M\}, m' \in \{1, \ldots, M\}$ **do**
12: $\quad\quad$ **if** $Y_X = \hat{Y}_X[m] \neq \hat{Y}_X[m']$ **then**
13: $\quad\quad\quad \mathbb{D} \leftarrow \mathbb{D} \cup \{(q_f, a_f[m]) \succ (q_f, a_f[m'])\}$ $\quad\quad\quad$ ▷ Elicit factual preferences
14: $\quad\quad$ **end if**
15: $\quad\quad$ **if** $Y_{X'} = \hat{Y}_{X'}[m] \neq \hat{Y}_{X'}[m']$ **then**
16: $\quad\quad\quad \mathbb{D} \leftarrow \mathbb{D} \cup \{(q_{cf}, a_{cf}[m]) \succ (q_{cf}, a_{cf}[m'])\}$ $\quad$ ▷ Elicit counterfactual preferences
17: $\quad\quad$ **end if**
18: $\quad$ **end for**
19: **end for**

---

**Algorithm 3** Preference-based Causal Consistency Feedback

---

1: **Inputs:** Demonstrations $\mathcal{D} = \{\mathcal{P}_{X_i \to Y_i}\}$, question templates $q, \tilde{q}$, answer extractor $h$, target model $\ell_0$
2: **Output:** Dataset $\mathbb{D} = \{(q_f, a_f, q_{cf}, a_{cf}) \succ (q_f', a_f', q_{cf}', a_{cf}')\}$ of preferences over dialogues

---

3:   $\mathbb{D} \leftarrow \{\}$
4: **for** $\mathcal{P}_{X_i \to Y_i} \in \mathcal{D}, n \in \{1, \ldots, N\}$ **do**
5:     $u, X, Y, Y_{X'} \sim \mathcal{P}_{X_i \to Y_i}$              $\triangleright$ Sample a context, cause, and potential effects
6:     $q_f \leftarrow q(u), q_{cf} \leftarrow \tilde{q}_{X'}(u)$             $\triangleright$ Generate questions given the context
7:     **for** $m \in \{1, \ldots, M\}$ **do**
8:         $a_f[m] \sim \ell_0(q_f), a_{cf}[m] \sim \ell_0(q_f, a_f[m], q_{cf})$     $\triangleright$ Collect answers from the language model
9:         $\hat{Y}_X[m] \leftarrow h(a_f[m]), \hat{y}_{X'}[m] \leftarrow h(a_{cf}[m])$     $\triangleright$ Extract outcome estimates from answers
10:        $r[m] \leftarrow \mathcal{R}(\hat{Y}_X[m], \hat{Y}_{X'}[m]; u), r[m'] \leftarrow \mathcal{R}(\hat{Y}_X[m'], \hat{Y}_{X'}[m']; u)$   $\triangleright$ Compute consistency scores
11:     **end for**
12:     **for** $m \in \{1, \ldots, M\}, m' \in \{1, \ldots, M\}$ **do**
13:        **if** $r[m] \geq r[m']$ **then**
14:           $\mathbb{D} \leftarrow \mathbb{D} \cup \{(q_f, a_f[m], q_{cf}, a_{cf}[m]) \succ (q_f, a_f[m'], q_{cf}, a_{cf}[m'])\}$     $\triangleright$ Elicit preferences
15:        **end if**
16:     **end for**
17: **end for**

---

# D  DETAILS OF THE EXPERIMENTS

In all our experiments, we specifically used the version of Phi-3 mini available on `https://huggingface.co/microsoft/Phi-3-mini-128k-instruct`. This version has a context length of 128k and 3.8B parameters, and has been fine-tuned for instruction following. Table 5 summarizes its performance on various reasoning benchmarks in comparison with other language models, as reported in the Hugging Face repository. Overall, Phi-3 mini demonstrates competitive performance on these benchmarks.

Table 5: *Reasoning performance of Phi-3-Mini-128K-Ins in comparison to other language models.*

| Benchmark | Phi-3-Mini-128K-Ins | Gemma-7B | Mistral-7B | Mixtral-8x7B | Llama-3-8B-Ins | GPT3.5-Turbo-1106 |
|---|---|---|---|---|---|---|
| ARC Challenge 10-shot | 85.5 | 78.3 | 78.6 | 87.3 | 82.8 | 87.4 |
| BoolQ 0-shot | 77.1 | 66 | 72.2 | 76.6 | 80.9 | 79.1 |
| MedQA 2-shot | 56.4 | 49.6 | 50 | 62.2 | 60.5 | 63.4 |
| OpenBookQA 10-shot | 78.8 | 78.6 | 79.8 | 85.8 | 82.6 | 86 |
| PIQA 5-shot | 80.1 | 78.1 | 77.7 | 86 | 75.7 | 86.6 |
| GPQA 0-shot | 29.7 | 2.9 | 15 | 6.9 | 32.4 | 29.9 |
| Social IQA 5-shot | 74.7 | 65.5 | 74.6 | 75.9 | 73.9 | 68.3 |
| TruthfulQA (MC2) 10-shot | 64.8 | 52.1 | 53 | 60.1 | 63.2 | 67.7 |
| WinoGrande 5-shot | 71.0 | 55.6 | 54.2 | 62 | 65 | 68.8 |

When collecting datasets, we always sample 100 contexts per causal relationship and generate 10 answers for each question per context (see Table 6). This is with the exception of Section 5.1 where we double the number of contexts for OnlyF and OnlyCF to end up with the same number question-answer examples as other methods, and with the exception of OnlyFx2 in Appendix B.

Table 6: *Summary of Healthcare, Engineering, and Math Benchmarking problems.* We report the number of context samples for each generalization mode *if* that generalization mode is available within the corresponding problem domain. While we sample 100 contexts for each causal relationship but some generalization modes (namely induction and deduction) involve training on more than one causal relationship.

| Generalization Mode | Healthcare | Engineering | Math Benchmarking | | |
|---|---|---|---|---|---|
| | | | $S \to T$ | $S \to R$ | $R \to T$ |
| In-Domain | 100 | 100 | 100 | 100 | 100 |
| Common-Cause | 100 | 100 | (n/a) | (n/a) | (n/a) |
| Common-Effect | 100 | 100 | (n/a) | (n/a) | (n/a) |
| Inductive | (n/a) | 200 | 200 | (n/a) | (n/a) |
| Deductive (Cause-Based) | 400 | (n/a) | (n/a) | (n/a) | 200 |
| Deductive (Effect-Based) | (n/a) | (n/a) | (n/a) | 200 | (n/a) |

In order to obtain error bars, we repeat each experiment five times. The extractor $h$ is implemented using Llama 3 8B with the following prompt:

> I will give you a question and its answer. Determine whether the meaning of the answer is 'POSITIVE' or 'NEGATIVE'. An answer is 'POSITIVE' if it contains phrases like 'yes', 'it holds', 'correct', 'true', or similar affirmations. An answer is 'NEGATIVE' if it contains phrases like 'no', 'it does not hold', 'incorrect', 'false', or similar negations. Respond only with one word: 'POSITIVE' or 'NEGATIVE'. Question: '{$q$}' Answer: '{$a$}' Is the meaning 'POSITIVE' or 'NEGATIVE'?

Similarly, the answer generator $H$, in the case of supervised counterfactual feedback, is implemented using Llama 3 8B with the following prompt:

> I will give you a question and the initial word of its answer. Complete the answer starting form the provided word. Respond only with the complete answer. Question: {$q$} Answer: {No/Yes}, ...

### D.1 REASONING PROBLEM: PUZZLE

This hand-crafted puzzle, centered around a candy party, has been used in the experiments conducted in Section 5.1 and 5.2. Based on different generalization modes, we developed three variations of this puzzle, each featuring distinct causal structures:

**Structure-1: Bipartite Graph**

- **Context:** Anna, Bill, Cory, and Dave are going to a party, where the host is going to distribute candies. Anna will be happy if she gets at least 4 candies. Bill will be happy if he gets at least 6 candies. Cory will be happy if Anna and Bill are both happy or if he gets at least 8 candies. Dave will be happy if Anna and Bill are both happy or if he gets at least 10 candies. After distributing the candies, Anna gets {$N_A$}, Bill gets {$N_B$}, Cory gets {$N_C$}, and Dave gets {$N_D$}.

- **Factual Question:** Is {Anna/Bill/Cory/Dave} happy? Be as concise as possible.

- **Interventional Question:** Now, suppose that {Anna/Bill/Cory/Dave} {is/is not} happy regardless of the candy distribution. With this assumption, is {Anna/Bill/Cory/Dave} happy? Be as concise as possible.

- **Causal Relationships:**

$$A = N_A \geq 4 \tag{12}$$
$$B = N_B \geq 6 \tag{13}$$
$$C = (A \wedge B) \vee (N_C \geq 8) \tag{14}$$
$$D = (A \wedge B) \vee (N_D \geq 10) \tag{15}$$

- **Causal Structure:**

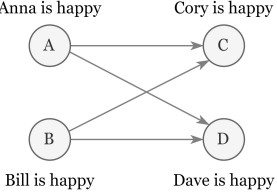

**Structure-2: Chain with No Direct Effect (NDE)**

- **Context:** Anna, Bill, and Cory are going to a party, where the host is going to distribute candies. Anna will be happy if she gets at least 5 candies. Bill will be happy if Anna is happy or if he gets at least 7 candies. Cory will be happy if Bill is happy or if he gets at least 9 candies. After distributing the candies, Anna gets {$N_A$}, Bill gets {$N_B$}, and Cory gets {$N_C$}.

- **Factual Question:** Is {Anna/Bill/Cory} happy? Be as concise as possible.

- **Interventional Question:** Now, suppose that {Anna/Bill/Cory} {is/is not} happy regardless of the candy distribution. With this assumption, is {Anna/Bill/Cory} happy? Be as concise as possible.

- **Causal Relationships:**

$$A = N_A \geq 5 \tag{16}$$
$$B = A \vee (N_B \geq 7) \tag{17}$$
$$C = B \vee (N_C \geq 9) \tag{18}$$

- **Causal Structure:**

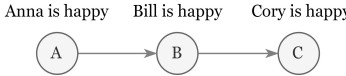

**Structure-3: Chain With Direct Effect (WDE)**

- **Context:** Anna, Bill, and Cory are going to a party, where the host is going to distribute candies. Anna will be happy if she gets at least 5 candies. Bill will be happy if Anna is happy or if he gets at least 7 candies. Cory will be happy if Annd and Bill are both happy or if he gets at least 9 candies. After distributing the candies, Anna gets {$N_A$}, Bill gets {$N_B$}, and Cory gets {$N_C$}.

- **Factual Question:** Is {Anna/Bill/Cory} happy? Be as concise as possible.

- **Interventional Question:** Now, suppose that {Anna/Bill/Cory} {is/is not} happy regardless of the candy distribution. With this assumption, is {Anna/Bill/Cory} happy? Be as concise as possible.

- **Causal Relationships:**

$$A = N_A \geq 5 \tag{19}$$
$$B = A \vee (N_B \geq 7) \tag{20}$$
$$C = (A \wedge B) \vee (N_C \geq 9) \tag{21}$$

- **Causal Structure:**

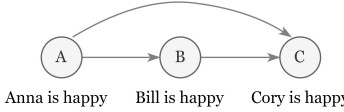

## D.2   REAL-WORLD REASONING PROBLEM: HEALTHCARE

In Section 5.3 we introduced three real-world problems to validate our experimental findings from the proof-of-concept puzzle reasoning problem. Here, we offer a more detailed explanation of one of these real-world reasoning problems—the **Healthcare** problem.

- **Context:** There are four types of breast cancer patients (based on their ERPR and HER2 indicators): (1) If a patient is ERPR positive and HER2 negative, they are 'Luminal A'. All luminal A patients should undergo surgery. (2) If a patient is ERPR positive and HER2 positive, they are 'Luminal B'. Luminal B patients should undergo surgery if their tumor is smaller than 1 cm and there is no nodal involvement. Luminal B patients should undergo therapy if their tumor is larger than 1 cm or if there is nodal involvement. (3) If a patient is ERPR negative and HER2 positive, they are 'Enriched'. Enriched patients should undergo surgery if their tumor is smaller than 1 cm and there is no nodal involvement. Enriched patients should undergo therapy only if their tumor is larger than 1 cm (even if there is nodal involvement). (4) If a patient is ERPR negative and HER2 negative, they are 'Basal'. Basal patients should undergo surgery if their tumor is smaller than 1 cm and there is no nodal involvement. Basal patients should undergo therapy only if their tumor is larger than 1 cm (even if there is nodal involvement). Jane is ERPR {negative/positive} and HER2 {negative/positive}. Her tumor is {$T_{cm}$} cm and there is {nodal involvement/no nodal involvement}.

- **Factual Question:** Will she undergo {surgery/therapy}? Be as concise as possible.

- **Possible Interventional Questions:** If {Jane had been ERPR positive/Jane had been ERPR negative/Jane had been HER2 positive/Jane had been HER2 negative/the tumor had been larger than 1 cm/the tumor had been smaller than 1 cm/there had been nodal involvement/there had been no nodal involvement}, would she have undergone {surgery/therapy}? Be as concise as possible.

- **Causal Relationships:**

$$H_{\text{ERPR}}, H_{\text{HER2}} \sim \begin{cases} (1,0) & \text{with probability } 0.50 \\ (1,1) & \text{with probability } 0.15 \\ (0,1) & \text{with probability } 0.20 \\ (0,0) & \text{with probability } 0.15 \end{cases} \tag{22}$$

$$C_{\text{type}} \sim \begin{cases} \text{Luminal A} & \text{if } H_{\text{ERPR}} \wedge \neg H_{\text{HER2}} \\ \text{Luminal B} & \text{if } H_{\text{ERPR}} \wedge H_{\text{HER2}} \\ \text{Enriched} & \text{if } \neg H_{\text{ERPR}} \wedge H_{\text{HER2}} \\ \text{Basal} & \text{if } \neg H_{\text{ERPR}} \wedge \neg H_{\text{HER2}} \end{cases} \tag{23}$$

$$T_{\text{cm}} \sim \begin{cases} \mathcal{N}(\mu = 3.07, \sigma = 2.22) & \text{if Luminal A} \\ \mathcal{N}(\mu = 2.96, \sigma = 1.45) & \text{if Luminal B} \\ \mathcal{N}(\mu = 2.42, \sigma = 1.03) & \text{if Enriched} \\ \mathcal{N}(\mu = 3.32, \sigma = 3.64) & \text{if Basal} \end{cases} \tag{24}$$

$$T = (T_{\text{cm}} \geq 1) \tag{25}$$

$$N \sim \begin{cases} \mathcal{B}(p = 86/251) & \text{if Luminal A} \\ \mathcal{B}(p = 35/79) & \text{if Luminal B} \\ \mathcal{B}(p = 18/32) & \text{if Enriched} \\ \mathcal{B}(p = 41/99) & \text{if Basal} \end{cases} \tag{26}$$

$$Y_{\text{surgery}} = \begin{cases} 1 & \text{if Luminal A} \\ \neg T \wedge \neg N & \text{if Luminal B} \\ \neg T \wedge \neg N & \text{otherwise} \end{cases} \tag{27}$$

$$Y_{\text{therapy}} = \begin{cases} 0 & \text{if Luminal A} \\ T \vee N & \text{if Luminal B} \\ T & \text{otherwise} \end{cases} \tag{28}$$

- **Causal Structure and In-domain Fine-tune/Evaluate Relations:**

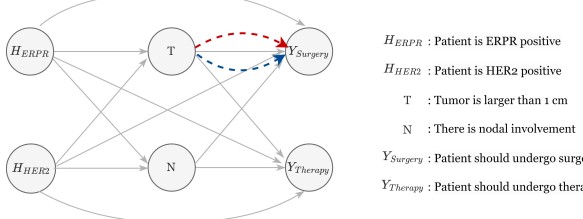

- **Causal Structure and Common-Cause Fine-tune/Evaluate Relations:**

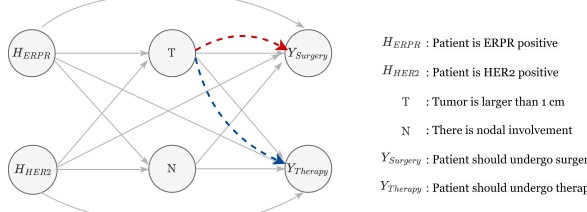

- **Causal Structure and Common-Effect Fine-tune/Evaluate Relations:**

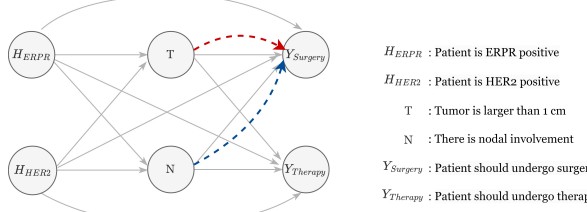

- **Causal Structure and Cause-Based Deduction Fine-tune/Evaluate Relations:**

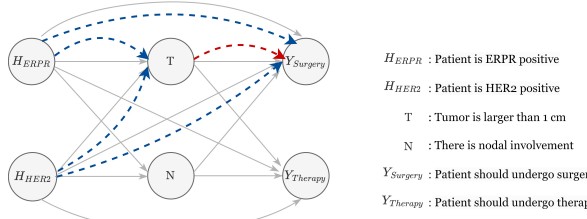

## D.3    REAL-WORLD REASONING PROBLEM: ENGINEERING

In Section 5.3 we introduced three real-world problems to validate our experimental findings from the proof-of-concept puzzle reasoning problem. Here, we offer a more detailed explanation of one of these real-world reasoning problems—the **Engineering** problem.

- **Context:** The type of fault on a transmission line is determined through three factors X, Y, and Z. These factors are 'close to zero' if they are less than 0.1. (1) If only one of the factors is close to zero, it is a line-to-line fault. When there is a line-to-line fault, it is BC fault if factor X is close to zero, AC fault if factor Y is close to zero, and AB fault if factor Z is close to zero. (2) If exactly two of the factors are close to zero, it is a line-to-ground fault. When there is a line-to-ground fault, it is AG fault if factors Y and Z are both close to zero, BG fault if factors X and Z are both close to zero, and CG fault if factors X and Y are both close to zero. For some faulty transmission line, X = X, Y = Y, and Z = Z.

- **Possible Factual Question:** {Is there a line-to-line/line-to-ground fault? / Is the fault type BC/AC/AB/AG/BG/CG?} Be as concise as possible.

- **Possible Interventional Questions:** If factor X/Y/Z had been/had not been close to zero, {would there have been a line-to-line/line-to-ground fault? / would the fault have been type BC/AC/AB/AG/BG/CG}? Be as concise as possible.

- **Causal Relationships:**

$$X \sim \mathcal{N}(\mu = \bar{X}, \sigma = 0.1) \tag{29}$$
$$Y \sim \mathcal{N}(\mu = \bar{Y}, \sigma = 0.1) \tag{30}$$
$$Z \sim \mathcal{N}(\mu = \bar{Z}, \sigma = 0.1) \tag{31}$$
$$X_0 = (X < 0.1) \tag{32}$$
$$Y_0 = (Y < 0.1) \tag{33}$$
$$Z_0 = (Z < 0.1) \tag{34}$$
$$LL = (X_0 \wedge \neg Y_0 \wedge \neg Z_0) \vee (\neg X_0 \wedge Y_0 \wedge \neg Z_0) \vee (\neg X_0 \wedge \neg Y_0 \wedge Z_0) \tag{35}$$
$$LG = (\neg X_0 \wedge Y_0 \wedge Z_0) \vee (X_0 \wedge \neg Y_0 \wedge Z_0) \vee (X_0 \wedge Y_0 \wedge \neg Z_0) \vee (X_0 \wedge Y_0 \wedge Z_0) \tag{36}$$
$$BC = LL \wedge X_0 \tag{37}$$
$$AC = LL \wedge Y_0 \tag{38}$$
$$AB = LL \wedge Z_0 \tag{39}$$

$$AG = LG \wedge Y_0 \wedge Z_0 \tag{40}$$
$$BG = LG \wedge X_0 \wedge Z_0 \tag{41}$$
$$CG = LG \wedge X_0 \wedge Y_0 \tag{42}$$

where $\bar{X}, \bar{Y}$, and $\bar{C}$ are drawn randomly from the values reported in Reddy et al. (2016).

- **Causal Structure and In-domain Fine-tune/Evaluate Relations:**

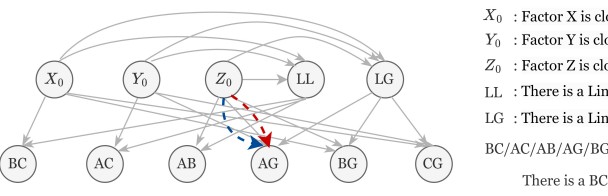

- **Causal Structure and Common-Cause Fine-tune/Evaluate Relations:**

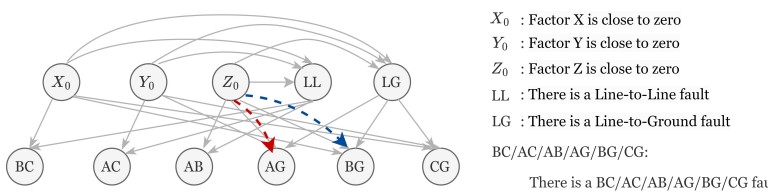

- **Causal Structure and Common-Effect Fine-tune/Evaluate Relations:**

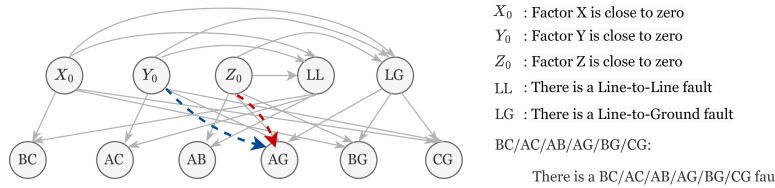

- **Causal Structure and Induction Fine-tune/Evaluate Relations:**

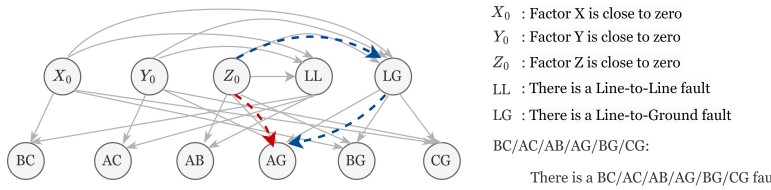

## D.4 REAL-WORLD REASONING PROBLEM: MATH BENCHMARKING

In Section 5.3 we introduced three real-world problems to validate our experimental findings from the proof-of-concept puzzle reasoning problem. Here, we offer a more detailed explanation of one of these real-world reasoning problems—the **Math Benchmarking** problem.

- **Context:** Carla is downloading a $\{N_{\text{size}}\}$ GB file. Normally she can download 2 GB/minute, but in 100 minutes, Windows will force a restart to install updates, which takes $\{N_{\text{minutes}}\}$ minutes. After the restart, Carla can resume her download.

- **Possible Factual Question:** {Will Windows force a restart before the download is complete? / Will the download take longer than 120 minutes?} Be as concise as possible.

- **Possible Interventional Questions:** If {she were downloading a file twice the size / Windows had forced a restart before the download was complete / Windows had not forced a restart before the download was complete}, would {Windows have forced a restart before the download was complete? / the download have taken longer than 120 minutes?} Be as concise as possible.

- **Causal Relationships:**

$$N_{size} \sim \mathcal{U}(50, 300) \tag{43}$$

$$N_{minutes} \sim \mathcal{U}(10, 30) \tag{44}$$

$$S \sim \mathcal{B}(p = 0.5) \tag{45}$$

$$N_{download\_time} = [N_{size} * 2 * S + N_{size}(1 - S)]/2 \tag{46}$$

$$R = (N_{download\_time} \geq 100) \tag{47}$$

$$T = (N_{download\_time} + R * N_{minutes}) \geq 120) \tag{48}$$

- **Causal Structure and Induction Fine-tune/Evaluate Relations:**

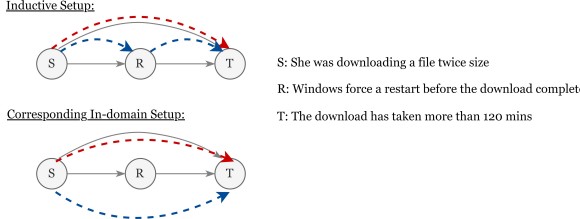

- **Causal Structure and Deduction Cause-Based Fine-tune/Evaluate Relations:**

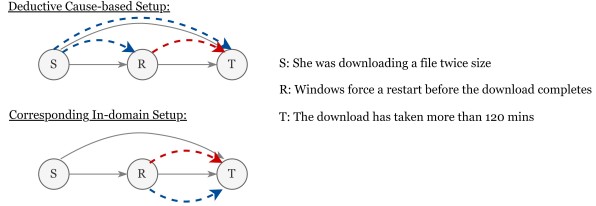

- **Causal Structure and Deduction Effect-Based Fine-tune/Evaluate Relations:**

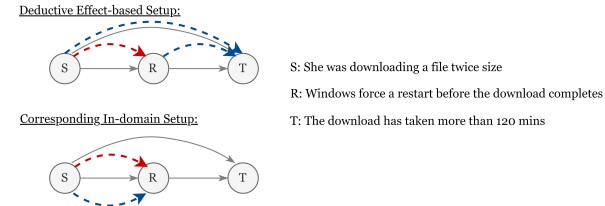

# E    METHODS BESIDES FINE-TUNING

## E.1    CHAIN-OF-THOUGHT PROMPTING

Responses of a language model can be improved in different ways, not just through fine-tuning. For instance, one could pursue prompt engineering instead. When eliciting reasoning, our focus has been fine-tuning while chain-of-thought prompting is a form of prompt engineering. These two approaches mainly differ in terms of what point they intervene on: Prompt engineering modifies the inputs passed to the language model whereas fine-tuning modifies the language model itself. Importantly, this means that the use of these two techniques are largely orthogonal to each other and definitely not mutually exclusive. In our case, chain-of-thought prompting is not a competing method against ours but rather a separate avenue for further improvement that could be pursued in conjunction with fine-tuning. Because of this, we did not consider it as a baseline in our experiments.

Notably, when used to elicit reasoning, chain-of-thought prompting would be subject to the same evaluation challenge that our fine-tuning approach faces: Examples provided in a chain-of-thought prompt might lead to more accurate answers for "in-domain" questions, however, this does not mean the examples was successful in eliciting reasoning unless the accuracy generalizes to new problems (that require forming novel chains using the same concepts).

## E.2 FORMAL REPRESENTATIONS AND DEDICATED SOLVERS

Since the reasoning problems we consider are straightforward to solve once translated into a formal representation (like the structural equations in Appendix D), an alternative solution strategy could be to extract such formal representations from the natural language descriptions of the problem and rely on dedicated solvers to compute outcomes in an exact manner. However, the extraction of formal representations would remain a bottleneck for performance in such a strategy as off-the-shelf language models are not effective at this task. For instance, OpenMathInstruct-1 (Toshniwal et al., 2024) achieves an accuracy of 84.6% on GSM8K by generating "code-interpreter" solutions, which list the computations performed to arrive at the final answer in a formal language. However, this required fine-tuning CodeLlama-70B to generate responses in the style of these code-interpreter solutions using a custom dataset with 18 million problem-to-code examples.

In short, although extracting formal representations (like code-interpreter solutions) could enable exact computation of answers thereafter, fine-tuning a language model to accurately extract such representations is likely a more complex task than fine-tuning the model to provide direct answers. This challenge is especially pronounced for small language models like Phi-3 mini, which has 3.8B parameters compared to the 70B of OpenMathInstruct-1. This makes our approach even more suited to the domain of small language models.

Additionally, our setting poses an even further complication for the formal-representation route: The reasoning problems we consider are not simple forward computations as in math problems but rather involve causal systems with multiple possible interventions, each intervention altering the forward computation necessary to obtain the final answer. In order to highlight all these challenges, we report a qualitative evaluation of Phi-3 mini in terms of its ability to extract formal representations of causal scenarios. We designed the following one-shot prompt using the problem in Figure 5 as an example. We then used this prompt to extract structural equations for our other problem domains: Healthcare, Engineering, and Math Benchmarking.

> I want you to generate structural equations from a causal scenario described in English. Respond only with the structural equations. Here is an example:
>
> Scenario:
> Anna, Bill, Cory, and Dave are going to a party, where the host is going to distribute candies. Anna will be happy if she gets at least 4 candies. Bill will be happy if he gets at least 6 candies. Cory will be happy if Anna and Bill are both happy or if he gets at least 8 candies. Dave will be happy if Anna and Bill are both happy or if he gets at least 10 candies.
>
> Structural Equations:
> NA: The number of candies Anna gets
> NB: The number of candies Bill gets
> NC: The number of candies Cory gets
> ND: The number of candies Dave gets
> HA: Happiness of Anna
> HB: Happiness of Bill
> HC: Happiness of Cory
> HD: Happiness of Dave
> HA = (NA >= 4)
> HB = (NB >= 6)
> HC = (HA and HB) or (NC >= 8)
> HD = (HA and HB) or (ND >= 10)
>
> Scenario:
> {Test Scenario}
> Structural Equations:

**Results for Healthcare.**

Test scenario:

> There are four types of breast cancer patients (based on their ERPR and HER2 indicators): (1) If a patient is ERPR positive and HER2 negative, they are 'Luminal A'. All luminal A patients should undergo surgery. (2) If a patient is ERPR positive and HER2 positive, they are 'Luminal B'. Luminal B patients should undergo surgery if their tumor is smaller than 1 cm and there is no nodal involvement. Luminal B patients should undergo therapy if their tumor is larger than 1 cm or if there is nodal involvement. (3) If a patient is ERPR negative and HER2 positive, they are 'Enriched'. Enriched patients should undergo surgery if their tumor is smaller than 1 cm and there is no nodal involvement. Enriched patients should undergo therapy only if their tumor is larger than 1 cm (even if there is nodal involvement). (4) If a patient is ERPR negative and HER2 negative, they are 'Basal'. Basal patients should undergo surgery if their tumor is smaller than 1 cm and there is no nodal involvement. Basal patients should undergo therapy only if their tumor is larger than 1 cm (even if there is nodal involvement).

Response of Phi-3 mini:

> NA: ERPR status of the patient
> NH: HER2 status of the patient
> TA: Tumor size of the patient
> TN: Nodal involvement of the patient
> SA: Surgery for the patient
> TH: Therapy for the patient
> LA: Luminal A patient
> LB: Luminal B patient
> EN: Enriched patient
> B: Basal patient
> SA = (NA = 'ERPR positive'and NH = 'HER2 negative'and TA $<$1 cm and TN = 'No nodal involvement')
> LB = (NA = 'ERPR positive'and NH = 'HER2 positive'and (TA $<$1 cm and TN = 'No nodal involvement'or TA $>$1 cm or TN = 'Nodal involvement'))
> EN = (NA = 'ERPR negative'and NH = 'HER2 positive'and (TA $<$1 cm and TN = 'No nodal involvement'or TA $>$1 cm))
> B = (NA = 'ERPR negative'and NH = 'HER2 negative'and (TA $<$1 cm and TN = 'No nodal involvement'or TA $>$1 cm))
> LA = (NA = 'ERPR positive'and NH = 'HER2 negative')
> LB = (NA = 'ERPR positive'and NH = 'HER2 positive')
> EN = (NA = 'ERPR negative'and NH = 'HER2 positive')
> B = (NA = 'ERPR negative'and NH = 'HER2 negative')
> SA = (LA or LB or EN or B)
> TH = (not SA)

Beyond whether it is accurate or not, this response lacks a clear formal interpretation, as many of the variables are defined multiple times (causing ambiguity).

**Results for Engineering.**

Test scenario:

> The type of fault on a transmission line is determined through three factors X, Y, and Z. These factors are 'close to zero' if they are less than 0.1. (1) If only one of the factors is close to zero, it is a line-to-line fault. When there is a line-to-line fault, it is BC fault if factor X is close to zero, AC fault if factor Y is close to zero, and AB fault if factor Z is close to zero. (2) If exactly two of the factors are close to zero, it is a line-to-ground fault. When there is a line-to-ground fault, it is AG fault if factors Y and Z are both close to zero, BG fault if factors X and Z are both close to zero, and CG fault if factors X and Y are both close to zero.

Response of Phi-3 mini:

> XC = (X <0.1) and (Y >= 0.1) and (Z >= 0.1)
> YC = (X >= 0.1) and (Y <0.1) and (Z >= 0.1)
> ZC = (X >= 0.1) and (Y >= 0.1) and (Z <0.1)
> BC_fault = XC
> AC_fault = YC
> AB_fault = ZC
> YZ_close = (Y <0.1) and (Z <0.1)
> XZ_close = (X <0.1) and (Z <0.1)
> XY_close = (X <0.1) and (Y <0.1)
> AG_fault = YZ_close
> BG_fault = XZ_close
> CG_fault = XY_close

This response is accurate except that it marginalizes out the general fault types, namely line-to-line and line-to-ground. While it is possible to compute factual values for specific fault types using these equations, the same is not true for the general fault types. Additionally, this limitation prevents us from performing any interventions on the general fault types.

**Results for Math Benchmarking.**

Test scenario:

> Carla is downloading a 250 GB file. Normally she can download 2 GB/minute, but in 100 minutes, Windows will force a restart to install updates, which takes 20 minutes. After the restart, Carla can resume her download. Consider how long it will take to download the file.

Response of Phi-3-Mini:

> NA: The amount of data Carla downloads before the restart
> NR: The amount of data Carla downloads after the restart
> T: Total time to download the file
> NA = 2 GB/minute * 100 minutes
> NR = (250 GB - NA) / 2 GB/minute
> T = 100 minutes + 20 minutes + NR/2 minutes

This response best underlines the additional challenges involved in extracting *causal* equations. First of all, the calculations provided in the response are not entirely accurate, even for the specific instance of the scenario represented. Setting aside these small inaccuracies, the response assumes that a restart will necessarily occur. With these equations, it is not possible to reason about outcomes under interventions such as "What if Windows had not forced a restart?" or "What if Carla had been downloading a file half the size?"—both of which are types of interventions considered in our experiments. Even if we were to modify these equations to explicitly include whether a restart occurs or not as a variable, it may not always be possible to anticipate what other kinds of interventions will be queried. For instance, these equations also fail to account for potential interventions on the download speed.

## F  DISAMBIGUATION OF "GENERALIZATION"

Generalizing from a few examples to a larger input space, potentially involving never-seen-before inputs as in the case of 2-4 digit multiplication to 5-7 digit multiplication, is a useful and non-trivial property for a language model to have. This type of generalization remains a challenge within what we term the "in-domain" problem. In fact, the results in Figure 5 show how DPO+CCF outperforms other baselines in this regard.

While there are many notions of generalization, what we argue is that *reasoning* fundamentally involves breaking a problem down to its basic components and synthesizing them in novel ways that enable generalization to entirely new problems. In a causal graph, this may correspond to an

understanding of individual edge relations in a way that would enable one to make estimations for any composite relationship. This is the type of generalization we are interested in. Here, sample efficiency in learning to solve a fixed problem would not necessarily be indicative of reasoning ability.

## G  An Intuitive Explanation of Our Inconsistency Rates

An alternative interpretation of the inconsistency rates we introduce is to frame reasoning about necessity or sufficiency as a classification problem. For instance, focusing on necessity, each context variable $U$, along with the cause $X$ and the potential effect $Y, Y_{X'}$ induced by $U$, needs to be assigned to one of three classes:

(i) $\mathbb{N}$: The cause and the effect both occurred together, and the cause was necessary for the effect to occur, meaning the effect would not have occurred otherwise if the cause was prevented. In a case like this, one might say "For the observed effect, the cause was a necessary condition."

(ii) $\mathbb{N}'$: The cause and the effect both occurred together, but the cause was not necessary for the effect to occur, meaning the effect would have occurred anyways even if the cause was prevented. In a case like this, one might say "For the observed effect, the cause was not a necessary condition."

(iii) $\emptyset$: It is not meaningful to talk about necessity as either the cause or the effect did not occur. Note that the previous cases take the occurrence of the cause and the effect as a given, and make statements about what would have happened to the effect if the cause were to be prevented. This is consistent with Pearl's definition of PN as a conditional probability with the condition being $X = x$ and $Y = y$.

According to the underlying causal model, each context has a corresponding ground-truth "label" as defined in Equation 6 Given a language model, its answers provide estimates for the unknown potential outcomes, which in turn imply a prediction of this ground-truth "label". The necessity inconsistency rate (N-IR) is the error rate for this classification task (how often the predicted label differs from the ground-truth label).

## H  Further Analysis of The Generalization Results

In this section, we expand upon our analysis in Section 5.2, giving more detailed explanations as well as share new observations.

In Figure 7a, we see that common-effect generalization leads to an asymmetric improvement in N-IR and S-IR. This is because, in common-effect generalization, the effect of interest is fixed between training and testing, hence the factual questions regarding this fixed effect remain the same as well (factual questions do not depend on the cause as there are no interventions involved). Meanwhile, the cause that is being intervened on changes, hence the fine-tuned model now needs to answer a different set of counterfactual questions during test time than the ones seen during training time. Therefore, we would expect common-effect generalization to be more challenging in terms of CF-ER as opposed to F-ER. Since in our constructed examples, causes tend to be sufficient for effects rather than necessary, performing well in terms of S-iR is more dependent on counterfactual question answering while N-IR is more dependent on factual question answering. Most contexts can be dismissed as irrelevant as far as necessity is concerned (i.e. $\mathcal{N} = \emptyset$) by estimating the factual outcome to be negative, whereas determining sufficiency requires determining whether the counterfactual outcome is positive or it is also negative. Therefore, we see a greater improvement in terms of N-IR.

Across all generalization modes presented in Figure 7, by far the most challenging one is cause-based deduction with direct effect ("WDE"), where no method strictly improves the base model. This is due to the problem structure: When $A$ affects $C$ both directly as well as indirectly through $B$, it is not possible to separate the two types of effects without seeing any interventions on the intermediary $B$. Notice how this stops being an issue for cause-based deduction with no direct effect ("NDE"). Interestingly, we still see an improvement in terms of N-IR (at the cost of worse performance in S-IR). This is similar to the previous example: Seeing demonstrations for $A \to C$ at least improves the model's ability to answer factual questions regarding $C$, which translates to better F-ER when reasoning about $B \to C$, which in turn translates to better N-IR as before ($B$ is sufficient for $C$ hence

most contexts can readily be dismissed as irrelevant as far as necessity is concerned just by estimating the factual outcome correctly).

Finally, when there is no direct effect, we see that cause-based deduction and effect-based deduction are similarly effective. However, cause-based deduction is slightly better in terms of S-IR whereas effect-based deduction is slightly better in terms of N-IR. This is again the result of each variable being sufficient for the next variable. When reasoning about $B \to C$ (cause-based deduction), determining sufficiency requires estimating the outcome under the intervention $B = 0 \to 1$ when $B = 0$. Since $A$ is sufficient for $B$, this intervention can indirectly be performed through the intervention $A = 0 \to 1$. However, determining sufficiency requires estimating the outcome under the intervention $B = 1 \to 0$ when $B = 1$. This can no longer be achieved consistently through $A$ (it could still be the case that $B = 1$ even after the intervention $A = 1 \to 0$). Therefore, S-IR is the easier metric for cause-based deduction. A similar thing happens when reasoning about $A \to B$ (effect-based deduction). Determining necessity requires observing $B = 1 \to 0$ if we perform the intervention $A = 1 \to 0$. After performing $A = 1 \to 0$, if we observe $C = 1 \to 0$ instead, we can still conclude this must be because $B = 1 \to 0$ as $B$ is sufficient for $C$ (i.e. $B \implies C$). However, determining sufficiency requires observing $B = 0 \to 1$ after the intervention $A = 0 \to 1$. Here, it can already be the case that $C = 1$ as $B$ is not a necessary condition for $B$, preventing us to reason about $B$ indirectly through $C$.

## I FURTHER DISCUSSION

### I.1 WHAT TO DO WHEN CAUSES AND EFFECTS ARE NON-BINARY?

The math benchmarking domain (detailed in Appendix D.4) is a good example of how non-binary causes and effects can be handled within our formulation. In this domain, we consider a particular math problem within the GSM8K dataset, which is about Carla downloading a file with a certain size (denote it with $N_X$) and how long it would take to download this file in minutes (denote it with $N_Y$). Note that both of these variables would be non-binary. As an effect, we consider whether the download takes longer than 120 minutes, that is $Y = 1$ if $N_Y > 120$ and $Y = 0$ otherwise. As an intervention, we consider what would have happened if Carla was downloading a file that is twice the size. For this, we defined $X$ such that $X = 1$ if the intervention has occurred and $X = 0$ if it has not; and the post-intervention file size is given by $N_X' = X \cdot (2N_X) + (1 - X) \cdot N_X$.

Generally, when interested in a non-binary outcome $N_Y$, one can consider binary events dependent on that outcome instead. Similarly, when interested in interventions on a non-binary target $N_X$, one can decide on a particular intervention and let the cause $X$ be the binary indicator of whether that intervention has occurred or not. We mainly focused on binary causes and effects because the concepts of necessity and sufficiency are the most meaningful for binary variables. However, it is worth mentioning there is also work that aims to extend these concepts to categorical variables (Li and Pearl, 2024).

### I.2 DEFINING FALSE NECESSITY AND FALSE SUFFICIENCY

We can breakdown necessity and sufficiency inconsistency rates in to finer components such as false necessity and false sufficiency. In our framework, false necessity and false sufficiency can be interpreted as the events $\hat{\mathcal{N}} = \mathbb{N} \wedge \mathcal{N} \neq \mathbb{N}$ and $\hat{\mathcal{S}} = \mathbb{S} \wedge \mathcal{S} \neq \mathbb{S}$ respectively. However, notice that both of these events are the same event as $\hat{Y} \neq Y \vee \hat{Y}_{X'} \neq Y_{X'}$, that is either a factual or a counterfactual error has been made. These failures are already covered by factual and counterfactual error rates.

Overall, the individual inconsistency rates we introduce, N-IR, S-IR, AN-IR, and AS-IR, together with the individual error rates, F-ER and CF-ER, are already a finer breakdown of aggregated metrics like Avg-IR and Avg-ER, both of which represent general accuracy (the former accounts for correlations between factual and counterfactual errors whereas the latter looks at them independently).

