# OpenReview forum: "Reasoning Elicitation in Language Models via Counterfactual Feedback"
_ICLR.cc/2025/Conference — ICLR 2025 Oral_

### Official Review · Reviewer_7FpA · 2024-11-03

**Soundness:** 2
**Presentation:** 3
**Contribution:** 2
**Rating:** 6
**Confidence:** 4

**Summary:**

This paper explores the potential of fine-tuning language models (LLMs) to improve their causal reasoning abilities, particularly through counterfactual feedback. Standard LLMs exhibit a discrepancy between their strong recall abilities and their limited reasoning abilities. The authors propose a framework that incorporates new metrics (necessity and sufficiency inconsistency rates) to assess reasoning more comprehensively. They explore fine-tuning methods, including supervised and preference-based approaches, to improve understanding of causal dependencies and generalization across tasks. The study finds that fine-tuning is more effective at improving reasoning in inductive scenarios than in deductive ones, suggesting different patterns of generalization in reasoning tasks. Also, the paper suggests that using factual and counterfactual reasoning examples together can lead to better reasoning abilities in models.

**Strengths:**

The authors present a framework for fine-tuning based on causal reasoning, which provides a structured approach to understanding how reasoning can generalize across different problems. They formally categorize this generalization into four distinct types: common-effect, common-cause, inductive, and deductive reasoning, which helps to better understand reasoning in LLMs. This classification provides a systematic way to analyze and apply causal reasoning to different tasks.

The authors propose new metrics-Necessity Inconsistency Rates (N-IR) and Sufficiency Inconsistency Rates (S-IR)-for evaluating reasoning in large language models (LLMs) based on causal principles. They also introduce absent necessity and absent sufficiency to capture cause-effect relationships beyond traditional necessity and sufficiency. These metrics provide a more detailed framework for evaluating causal reasoning in LLMs.

The authors fine-tuned the models by including the counterfactual information with the factual information, which proved useful in improving the model's understanding of causal dependencies. This focus is consistent with human reasoning processes, where understanding "what-if" scenarios often plays a critical role in causal inference. Preference training using DPO was also used in the paper, which showed that learning from comparative feedback can be more effective than standard supervised approaches in some cases.

**Weaknesses:**

My main concern with the paper relates to the way the experiments are conducted. The paper claims that using factual and counterfactual examples leads to a performance gain compared to the base model or the model trained only on factual examples of counterfactual examples. However, this could be due to the increased number of training examples, which leads to an increase in overall performance.
In addition, a model tuned with both factual and counterfactual examples tends to perform well on metrics that measure both properties compared to either one. This may simply be due to an over-fitting of causal specific patterns. In addition, the experiments are limited to one model and one synthetically generated dataset, and the results are difficult to generalize across models and datasets. The other datasets used later are missing all the training/test splits used and how the model was trained. Section 6.3 is hard to understand because it lacks information on the number of samples used.

**Actionable Suggestion** : It would be great to see an experiment where you randomly sample equal number of factual and counterfactual examples and fine-tune the model and compare it with the individual factual and counterfactual only models. That would be a fair comparison.

The experiments also lack deeper analysis to better understand where fine-tuning fails. For example, a breakdown of errors by task type, complexity, or specific reasoning challenge (e.g., false sufficiency vs. false necessity errors) would provide insight into model weaknesses. The errors for which the model struggles to perform well need to be analyzed, and this is not discussed in the paper.

**Actionable Suggestion:** Please provide the task specific error categories that provides deeper insights into the models' strengths and weaknesses.

Finally, the approach of using counterfactual reasoning to improve model performance in reasoning tasks is not new and has been studied previously by https://arxiv.org/abs/2307.02477 and https://aclanthology.org/2022.lrec-1.229/.
The novelty of this paper lies in the way the two metrics are proposed to understand causal reasoning in LLMs. However, due to the weaker experiments, it is not clear why this paper stands out. It is similar in many ways to papers suggesting that more data can improve reasoning (such as from https://arxiv.org/abs/2309.12284).

**Questions:**

Section 6.3 lacks any information about the real-world data used. It is important to discuss the training and test data used and how many samples were rejected due to incorrect or unclear counterfactuals. It is mentioned at the end of the paper *Note that not all generalization modes were tested for each problem due to differences in causal structures, so the average scores were calculated using only the problems that were tested for each generalization mode.* which needs to be discussed.
**Provide a table summarizing the datasets used, including sample sizes, data sources, and which generalization modes were tested for each problem.** This will clear the confusion in this section.

Authors need to provide a discussion section where the type of error each type of fine-tuned model makes is discussed. Currently, just from the numbers, it is hard to understand what are the errors made by each model and how preference tuning can solve some of them. **Some error analysis examples or visualizations that would illustrate the different types of errors made by each model and how preference tuning addresses them would be helpful here**.

Some results need to be discussed more like statements like: *However, with common-cause/effect and deductive demonstrations, they no longer show the same reasoning improvements as observed in the in-domain setting.*  Inductive vs. Deductive results are not clear and need to be discussed why one approach performs better than the other.

Finally, the most important result to discuss is the data imbalance between the factual or counterfactual training set alone and a combination of the two. Both experiments need to be balanced for a fair comparison, otherwise it could just be the additional data points that lead to improved performance.

---

> ### Author Response · Authors · 2024-11-22
> **Response to Reviewer 7FpA (Comment 1/3)**
>
> Thank you for your thoughtful comments and suggestions! We give answers to each in turn, as well as pointing out corresponding updates we have made to the paper (changes are highlighted in blue in the revised paper).
>
> ---
>
> **(1) Balance of Training Data between Different Methods**
>
> For the results in Figure 5, OnlyF and OnlyCF has the same number of training examples as F&CF. We ensure this by sampling twice the number of context variables $U$ for OnlyF and OnlyCF compared with F&CF. In other words, F&CF has access to 100 context samples with two question-answer examples for each context sample (one factual, one counterfactual). Meanwhile, OnlyF and OnlyCF have access to 200 context samples with a single question-answer example for each context (either factual or counterfactual). So, the experimental setup for Figure 5 already matches the suggested setup in the review.
>
> **Update:** We now mention this detail regarding the experimental setup at the end of “Experimental Setup” in Section 6.1 (now 5.1 in the revised version) as well as in Appendix C.
>
> However, the experimental setup for the results in Table 1/2 is different. There, we sampled the same number of context variables for each method. This means F&CF has more question-answer examples in its training dataset, but crucially, each method still sees the same number of problem instances. For example, in the healthcare domain, OnlyF and F&CF sees the same set of breast cancer patients (each patient corresponding to a context); the training data for F&CF does not include any new patients; and the additional CF feedback is provided only for the existing patients. Therefore, the comparison between OnlyF and F&CF remains fair. Note that this would not have been the case if the context variables for the factual dataset and the counterfactual dataset were sampled independently (which is *not* the case in our experiments).
>
> In Figure 5, we preferred the setup with the same number of question-answer examples across methods to be absolutely certain that any improvement we see is due to the type of feedback being provided (like the reviewer has rightly raised as a concern). In Table 1/2, we preferred the setup with the same number of context variables across methods because this is likely to be the case in practice (for instance, if we apply these methods in a healthcare setting, the number of patients would be constant for each method).
>
> **Update:** We have now included this discussion regarding the two different approaches one could take when benchmarking OnlyF and F&CF in Appendix C.

---

> > ### Comment · Reviewer_7FpA · 2024-11-25
> > **Response**
> >
> > Thank you for the response.
> >
> > Mentioning that in Table 1/2, the training examples were kept same but the number of samples increased by 2X still looks like an unfair comparison. In my view, it is similar to data augmentation where both sides of the problem (F and CF) are shown to the model. Data augmentation has been proven to improve performance in many cases and this might be the case here. I know that for individual training of F and CF, more data cannot be generated but then the number of combined C and CF data can be reduced by half to make it fair. Otherwise, I am not convinced that data increase is not the reason for the improved performance.

---

> > > ### Author Response · Authors · 2024-11-27
> > > **Re: Response**
> > >
> > > Thank you for engaging with our response and voicing your remaining concerns!
> > >
> > > While we believe that a fair comparison is when all methods have access to the same number of context samples, we appreciate that having consistency between Figure 5 and Table 1/2 is important for the comparability of these two sets of results. To address this, we conducted additional experiments for the "In-Domain" portion of Table 1/2 so that OnlyF methods have access to twice the number of context variables (hence the same number of question-answer examples as F&CF, as in Figure 5). We call these new benchmarks *OnlyFx2* to differentiate them from the original setup.
> > >
> > > Results are provided below, where new entries are highlighted in bold. As expected, OnlyFx2 outperforms OnlyF in terms of both Avg-ER and Avg-IR. However, it still does not match the performance of F&CF, despite having access to a more diverse set of context variables and the same number of question-answer examples. A detailed breakdown of the new results is included further below (similar to Table 2). For the final version of the paper, we will complete these experiments for the remaining modes of generalization. Unfortunately, this is not feasible to do within the rebuttal period due to the time-intensive nature of the experiments (for instance, we have started running the in-domain portion on Nov 14).
> > >
> > > ---
> > >
> > > | Mode | Metric | Base | SFT+OnlyF | DPO+OnlyF | **SFT+OnlyFx2** | **DPO+OnlyFx2** | SFT+F&CF | DPO+F&CF | DPO+CCF |
> > > | --- | --- | --- | --- | --- | --- | --- | --- | --- | --- |
> > > | In-Domain | Avg-ER | 1.00 | 0.82 | 0.86 | **0.74** | **0.76** | 0.42 | 0.53 | 0.48 |
> > > | In-Domain | Avg-IR | 1.00 | 0.73 | 0.72 | **0.65** | **0.66** | 0.44 | 0.51 | 0.47 |
> > >
> > > ---
> > >
> > > | Scenario | Alg. | F-ER | CF-ER | Avg-ER | N-IR | S-IR | AN-IR | AS-IR | Avg-IR |
> > > | --- | --- | --- | --- | --- | --- | --- | --- | --- | --- |
> > > | Healthcare (In-domain) | SFT+OnlyFx2 | 3.09(0.01) | 21.41(0.01) | 12.25(0.01) | 2.39(0.01) | 1.15(0.00) | 2.82(0.01) | 21.06(0.01) | 6.86(0.01) |
> > > | Healthcare (In-domain) | DPO+OnlyFx2 | 5.18(0.32) | 24.70(0.61) | 14.94(0.45) | 4.21(0.40) | 2.75(0.05) | 5.09(0.28) | 21.68(0.13) | 8.43(0.19) |
> > > | Engineering (In-domain) | SFT+OnlyFx2 | 0.44(0.00) | 31.65(0.02) | 16.05(0.00) | 5.59(0.01) | 26.40(0.00) | 0.07(0.00) | 0.40(0.00) | 8.11(0.00) |
> > > | Engineering (In-domain) | DPO+OnlyFx2 | 4.21(0.06) | 30.13(0.00) | 17.17(0.01) | 5.54(0.00) | 26.77(0.01) | 0.92(0.03) | 3.49(0.02) | 9.18(0.01) |
> > > | Math $S\to T$ (In-domain) | SFT+OnlyFx2 | 26.59(0.38) | 26.57(0.01) | 26.58(0.13) | 0.00(0.00) | 42.88(0.31) | 28.33(0.47) | 0.00(0.00) | 17.80(0.10) |
> > > | Math $S\to T$ (In-domain) | DPO+OnlyFx2 | 24.47(0.41) | 48.01(1.30) | 36.24(0.61) | 0.00(0.00) | 59.29(0.82) | 28.08(0.39) | 0.00(0.00) | 21.84(0.14) |
> > > | Math $S\to R$ (In-domain) | SFT+OnlyFx2 | 35.95(0.36) | 56.36(0.08) | 46.15(0.16) | 0.00(0.00) | 61.39(0.16) | 45.74(0.34) | 0.00(0.00) | 26.78(0.06) |
> > > | Math $S\to R$ (In-domain) | DPO+OnlyFx2 | 39.06(0.84) | 60.64(1.60) | 49.85(0.56) | 0.00(0.00) | 69.24(0.39) | 44.19(1.08) | 0.00(0.00) | 28.36(0.09) |
> > > | Math $R\to T$ (In-domain) | SFT+OnlyFx2 | 27.80(1.14) | 64.03(0.05) | 45.92(0.38) | 19.43(0.17) | 51.94(0.03) | 20.63(0.24) | 7.58(0.36) | 24.90(0.17) |
> > > | Math $R\to T$ (In-domain) | DPO+OnlyFx2 | 24.47(0.41) | 24.54(0.18) | 24.50(0.08) | 29.80(0.21) | 7.40(0.44) | 1.74(0.06) | 23.25(0.64) | 15.55(0.04) |

---

> > > > ### Comment · Reviewer_7FpA · 2024-12-03
> > > >
> > > > Can you explain from where the dataset is coming from in this:
> > > >
> > > >  >To address this, we conducted additional experiments for the "In-Domain" portion of Table 1/2 so that OnlyF methods have access to twice the number of context variables (hence the same number of question-answer examples as F&CF, as in Figure 5). We call these new benchmarks OnlyFx2 to differentiate them from the original setup.
> > > >
> > > > I am still not convinced that this is a fair comparison as I would like the F&CF data to be halved rather than doubling the F data.
> > > > I will maintain my score as of now but can go up by a point if I am convinced that OnlyFx2 is the same comparison as halving the F&CF with Fx1, which I am not sure at this point.

---

> > > > > ### Author Response · Authors · 2024-12-03
> > > > >
> > > > > All our problem settings are synthetic simulations (they are based on real-world statistics but do not involve any real-world datasets), meaning we can sample as much data as we want in each problem setting. All the data-generating processes are provided in Appendix C. For instance, the healthcare datasets are generated according to Equations 22-28. Doubling data for OnlyF means taking twice as many samples according to these equations.
> > > > >
> > > > > More specifically,
> > > > >
> > > > > * For F&CF, for each causal relationship in the training set, we took N=100 samples from these data-generating processes, and for each sample, generated one factual and one counterfactual question-answer pair for a total of 2N question-answer pairs.
> > > > >
> > > > > * For OnlyFx1, we took N=100 samples, but generated only the factual question-answer pair for a total of N question-answer pairs.
> > > > >
> > > > > * For OnlyFx2, we took 2N=200 samples, so that when we generate only the factual question-answer pairs, we still end up with 2N question-answer pairs, the same number as F&CF!
> > > > >
> > > > > Halving the F&CF samples and comparing it with OnlyFx1 would have achieved the same. We would have taken N/2=50 samples, and generate one factual and one counterfactual question-answer pair, resulting in a total of N question-answer pairs (same as OnlyFx1).
> > > > >
> > > > > ---
> > > > >
> > > > > In short, "F&CF vs. OnlyFx2" and "Halved F&CF vs. OnlyFx1" are the same comparison. Only difference is the dataset sizes: In the former, each method receives twice as many question-answer pairs compared with the latter. Note that we are not using any real-world datasets, all our problem settings are simulations (based on real-world statistics). That is why we were able to easily sample more context variables for OnlyFx2. As we are not limited by data availability, we preferred the comparison with more data for lower variance in the results.
> > > > >
> > > > > Please let us know if this clarifies where the additional data is coming from, and please let us know if you have any additional concerns.

---

> > > > > > ### Comment · Reviewer_7FpA · 2024-12-03
> > > > > >
> > > > > > Thanks for the clarification.
> > > > > > It makes sense. I have raised my score to 6.

---

> ### Author Response · Authors · 2024-11-22
> **Response to Reviewer 7FpA (Comment 2/3)**
>
> **(2) Detailed Analysis of the Results**
>
> We would like to highlight that our analysis already covers different failure modes of fine-tuning for reasoning, in particular it forms connections between these failures and error types as well as problem structures. We emphasize two examples:
>
> 1. In Figure 6a, we see that common-effect generalization leads to an asymmetric improvement in N-IR and S-IR. This is because, in common-effect generalization, the effect of interest is fixed between training and testing, hence the factual questions regarding this fixed effect remain the same as well (factual questions do not depend on the cause as there are no interventions involved). Meanwhile, the cause that is being intervened on changes, hence the fine-tuned model now needs to answer a different set of counterfactual questions during test time than the ones seen during training time. Therefore, we would expect common-effect generalization to be more challenging in terms of CF-ER as opposed to F-ER. Since in our constructed examples, causes tend to be sufficient for effects rather than necessary, performing well in terms of S-iR is more dependent on counterfactual question answering while N-IR is more dependent on factual question answering. Most contexts can be dismissed as irrelevant as far as necessity is concerned (i.e. $\mathcal{N}=\emptyset$) by estimating the factual outcome to be negative, whereas determining sufficiency requires determining whether the counterfactual outcome is positive or it is also negative. Therefore, we see a greater improvement in terms of N-IR.
>
> 2. Across all generalization modes presented in Figure 6, by far the most challenging one is cause-based deduction with direct effect (“WDE”), where no method strictly improves the base model. This is due to the problem structure: When $A$ affects $C$ both directly as well as indirectly through $B$, it is not possible to separate the two types of effects without seeing any interventions on the intermediary $B$. Notice how this stops being an issue for cause-based deduction with no direct effect (“NDE”). Interestingly, we still see an improvement in terms of N-IR (at the cost of worse performance in S-IR). This is similar to the previous example: Seeing demonstrations for $A\to C$ at least improves the model's ability to answer factual questions regarding $C$, which translates to better F-ER when reasoning about $B\to C$, which in turn translates to better N-IR as before ($B$ is sufficient for $C$ hence most contexts can readily be dismissed as irrelevant as far as necessity is concerned just by estimating the factual outcome correctly).
>
> **Update:** We have now added a more detailed analysis of the generalization results in the newly added Appendix G, which includes the discussion above as well as additional observations. In particular, we explain why cause-based deduction, under no direct effect (“NDE”), results in larger improvement in S-IR while effect-based deduction leads to larger improvement in N-IR. In short, this is a consequence of $A$ being sufficient for $B$ and $B$ being sufficient for $C$ in our constructed example, illustrating how the underlying problem structure can influence the effectiveness of different modes of reasoning generalization.
>
> ---
>
> **(3) False Necessity and False Sufficiency**
>
> Breakdown of necessity and sufficiency inconsistency rates into finer components such as false necessity and false sufficiency is a very interesting idea. For instance, in our framework, false necessity and false sufficiency can be interpreted as the events $\hat{\mathcal{N}}=\mathbb{N}\wedge\mathcal{N}\neq\mathbb{N}$ and $\hat{\mathcal{S}}=\mathbb{S}\wedge\mathcal{S}\neq\mathbb{S}$ respectively. However, notice that both of these events are the same event as $\hat{Y}\neq Y\vee\hat{Y}\_{X’}\neq Y\_{X’}$, that is either a factual or a counterfactual error has been made. These failures are already covered by factual and counterfactual error rates.
>
> Overall, the individual inconsistency rates we introduce, N-IR, S-IR, AN-IR, and AS-IR, together with the individual error rates, F-ER and CF-ER, are already a finer breakdown of aggregated metrics like Avg-IR and Avg-ER, both of which represent general accuracy (the former accounts for correlations between factual and counterfactual errors whereas the latter looks at them independently). They offer a more nuanced insight into fine-tuning for reasoning as exemplified above.
>
> **Update:** We have now included this discussion in the newly added Appendix H.2.

---

> ### Author Response · Authors · 2024-11-22
> **Response to Reviewer 7FpA (Comment 3/3)**
>
> **(4) Previous Work and Novelty**
>
> The related work mentioned in the review aims to *evaluate* reasoning capabilities of language models. Meanwhile, our primary goal is to *elicit* reasoning, that is to tune language models to directly optimize their reasoning ability. As emphasized in our related work, this is the novel aspect of our work. Achieving this goal first requires us to define a precise optimization target. We do this by defining various inconsistency rates and generalization modes. On top of these contributions, we also propose novel algorithms for generating fine-tuning data that targets the metrics we formulate.
>
> We disagree that the conclusion of our paper can be reduced to “more data can improve reasoning.” In fact, we keep the size of the available training data constant in all our experiments. Instead, we investigate the type of data provided (factual vs. counterfactual) and the form in which it is provided (independently as in F&CF vs. pairwise as in CCF). Our conclusion is that having both factual and counterfactual examples during fine-tuning is essential, and that pairing factual and counterfactual examples related through a shared context can lead to further improvements.
>
> **Update:** We have now included Wu et al. (2023) and Frohberg & Binder (2022) as examples of reasoning evaluation in our related work. We have also clarified our conclusion at the end of the paper.
>
> ---
>
> **(5) Details of Real-World Problems**
>
> Details regarding the real-world problems in Section 6.3 are provided in Appendix C. We would like to clarify two points: (i) Although the problems in Section 6.3 are based on real-world processes and statistics, they are still fully simulated environments and do not involve the use of real-world datasets. (ii) Generalization modes that can be tested for each problem are fundamentally restricted by the causal graph underlying the problem. For instance, if a problem does not feature two variables colliding, then it would not be possible to test common-cause and common-effect generalization for that problem. All the generalization modes we tested for each problem domain are listed in the “Scenario” column of Table 2 (which is a detailed breakdown of Table 1 in Section 6.3).
>
> **Update:** We have now added Table 3, which summarizes all of our problem settings, available generalization modes for each setting, and the number of context variables sampled for each generalization mode in one place. Thank you for this suggestion!

---

### Official Review · Reviewer_dEY8 · 2024-11-03

**Soundness:** 4
**Presentation:** 4
**Contribution:** 3
**Rating:** 8
**Confidence:** 3

**Summary:**

This paper presents a method for fine-tuning large language models by means of counterfactual feedback. The authors present four categories of causal reasoning (common-effect, common-cause, inductive, and deductive). They also propose two new metrics to measure necessity and sufficiency under a reasoning framework (these metrics measure necessity and sufficiency inconsistency rates), by first presenting an example justification of these metrics. They then introduce three methods for fine-tuning LLMs with counterfactual feedback: supervised, preference-based and causal consistency feedback.

**Strengths:**

I see a lot of strengths in this paper, including:

- The paper presents an interesting framework for fine-tuning LLMs with counterfactual feedback, which is highly relevant and original.
- The quality of the paper is very good: it is very well structured, organized, and it integrates a multiplicity of concepts, scenarios and metrics for the problem at hand. In this sense, the paper is beyond complete.
- The significance of the paper appears to be very relevant for researchers in causal reasoning (with or without LLMs).

**Weaknesses:**

I see very few weaknesses in this paper. Here is a couple of things I would improve:

- Organization of the paper: the placement of related work is a bit odd - perhaps moving it to the end of the paper would help.
- Conclusion: the limitations in the conclusion feel a bit 'incomplete', in the sense that these leave room open for further research of non-binary representations of causes and effects. Can you give some hints on how the proposed method could be adapted in this case?
- There is a typo in line 144 of the paper: "interesting" --> "interested"

**Questions:**

- Can you give some hints on how the proposed method could be adapted to the case of non-binary representations of causes and effects?

---

> ### Author Response · Authors · 2024-11-22
> **Response to Reviewer dEY8**
>
> Thank you for your thoughtful comments and suggestions! We give answers to each in turn, as well as pointing out corresponding updates we have made to the paper (changes are highlighted in blue in the revised paper).
>
> ---
>
> **(1) Non-binary Causes and Effects**
>
> The math benchmarking domain (detailed in Appendix C.4) is a good example of how non-binary causes and effects can be handled within our formulation. In this domain, we consider a particular math problem within the GSM8K dataset, which is about Carla downloading a file with a certain size (denote it with $N_X$) and how long it would take to download this file in minutes (denote it with $N_Y$). Note that both of these variables would be non-binary. As an effect, we consider whether the download takes longer than 120 minutes, that is $Y=1$ if $N\_Y>120$ and $Y=0$ otherwise. As an intervention, we consider what would have happened if Carla was downloading a file that is twice the size. For this, we defined $X$ such that $X=1$ if the intervention has occurred and $X=0$ if it has not; and the post-intervention file size is given by $N'\_X = X\cdot (2 N\_X) + (1 - X) \cdot N_X$.
>
> Generally, when interested in a non-binary outcome $N_Y$, one can consider binary events dependent on that outcome instead. Similarly, when interested in interventions on a non-binary target $N_X$, one can decide on a particular intervention and let the cause $X$ be the binary indicator of whether that intervention has occurred or not. We mainly focused on binary causes and effects because the concepts of necessity and sufficiency are the most meaningful for binary variables.  However, it is worth mentioning there is also work that aims to extend these concepts to categorical variables (Li & Pearl, 2022, https://arxiv.org/abs/2208.09568).
>
> **Update:** We have now included this discussion in the newly added Appendix H.1.
>
> ---
>
> **(2) Organization of the Paper**
>
> **Update:** We have now moved the related work to the end of the paper, just before the conclusion. A more direct transition from the description of our algorithms to the experiments can be easier to follow. We thank the reviewer for this suggestion! We have also fixed the typo mentioned in the review.

---

> ### Comment · Reviewer_dEY8 · 2024-11-26
> **Keeping my score**
>
> I am happy with the answers to my questions and will keep my score.

---

### Official Review · Reviewer_fqJm · 2024-11-04

**Soundness:** 3
**Presentation:** 3
**Contribution:** 2
**Rating:** 5
**Confidence:** 4

**Summary:**

This paper is interested in LLM's ability of doing causal reasoning (that follows Pearl's counterfactual methods). For this they propose novel metrics that balance accuracy in factual and counterfactual questions and then propose several fine tuning approaches and evaluate them using their proposed metrics with respect to several examples. They show that their fine tuning methods lead to better performance with respect to their proposed metrics.

**Strengths:**

Causal reasoning is an important problem and Pearl's approach of using counterfactuals to do causal reasoning is sound and well studied. There is indeed necessity of more fine grained metrics when evaluating reasoning ability of LLMs and the papers efforts towards that is a plus.
The paper presents several fine tuning techniques and evaluate them using their proposed metrics with respect to several examples. The evaluations show that their fine tuning methods lead to better performance with respect to their proposed metrics.

**Weaknesses:**

The examples that are studied in the paper are very direct in terms of language and LLMs can translate them to a formal representation (LLMs are good at that) and then call a dedicated solver and thus achieve near 100% accuracy. What is the motivation then of doing it the way that is proposed in the paper?

Why is your approach preferable to using a formal representation and a dedicated solver?

That motivation must match with the way the examples and evaluation data are presented and studied.

**Questions:**

1. See the question in the weakness part.

2. Are the examples that are studied in the paper original to this paper?  (If so, they are a good contribution.) Or are they published in some other papers cited in this paper.

3. In the Introduction you say: "When the goal of fine-tuning is specifically to improve reasoning, a unique problem arises in evaluating the fine-tuned LLMs: we cannot just measure performance for a held-out set of test samples within the same reasoning task. If we do, it would be impossible to tell whether the LLM actually learned to reason or whether it is still recalling the demonstrations we have made during fine-tuning.*2* "

*2* For instance, chain-of-thought prompting aims to improve reasoning by providing examples of how a problem can be solved in smaller steps. While such prompting is effective, its effectiveness can be attributed to successful imitation of the provided examples and is not necessarily the result of true reasoning (Wei et al., 2022).

In Page 3 you say: "Modes of Generalization. As we have discussed in the introduction, an in-domain evaluation is not sufficient alone to assess the success of fine-tuning for reasoning."

I am a bit confused by these. Consider an analogy. Say I have examples of how to multiple
2, 3, and 4 digit numbers and using that a model learns to be able to multiple 5, 6, 7 ... digit numbers. Is that a good generalization?

Similarly your model may learn from a few examples and if it is able to address causality in new examples with a much more complicated graph, and more variables, will that not be a good generalization?

Kindly clarify the distinction you are making between in-domain evaluation and generalization, and how your approach differs from the type of generalization mentioned above.

---

> ### Author Response · Authors · 2024-11-22
> **Response to Reviewer fqJm**
>
> Thank you for your thoughtful comments and suggestions! We give answers to each in turn, as well as pointing out corresponding updates we have made to the paper (changes are highlighted in blue in the revised paper).
>
> ---
>
> **(1) Formal Representations and Dedicated Solvers**
>
> While the reasoning problems we consider are straightforward to solve once translated into a formal representation (like the structural equations in Appendix C), extracting such formal representations from the natural language descriptions of the problems remains a significant bottleneck for performance. Off-the-shelf language models are not effective at this task. For instance, OpenMathInstruct-1 (https://arxiv.org/pdf/2402.10176) achieves an accuracy of 84.6% on GSM8K (competitive but still far from 100% accuracy) by generating “code-interpreter” solutions, which list the computations performed to arrive at the final answer in a formal language. However, this required fine-tuning CodeLlama-70B to generate responses in the style of these code-interpreter solutions using a custom dataset with 18 million problem-to-code examples.
>
> In short, extracting formal representations (like code-interpreter solutions) could indeed enable exact computation of answers thereafter *but* fine-tuning a language model to accurately extract such representations is likely a more complex task than fine-tuning the model to provide direct answers. This challenge is especially pronounced for small language models like Phi-3-Mini, which has 3.8B parameters compared to the 70B of OpenMathInstruct-1. Given all this, our approach is well motivated, particularly within the domain of small language models.
>
> Additionally, our setting poses an even further complication for the formal-representation route: The reasoning problems we consider are not simple forward computations as in math problems but rather involve causal systems with multiple possible interventions, each intervention altering the forward computation necessary to obtain the final answer.
>
> **Update:** We have now included this discussion in a newly added Appendix D.2.
>
> ---
>
> **(2) Originality of the Reasoning Problems Studied**
>
> With the exception of the math benchmarking domain, which is based on the GSM8K dataset, all of the reasoning problems in our paper are original contributions. Even the math benchmarking domain required significant modifications to the original GSM8K problems to adapt them to a causal framework. Specifically:
>
> 1. The logic problems in Figures 5 and 6 were purpose-built to explore the different generalization modes we define in Section 2.
>
> 2. The healthcare domain is based on a real-life breast cancer treatment guideline. We converted this guideline into a causal model using statistics gathered from various medical journals (details provided in Appendix C.2) and crafted prompts that align with this causal model (also detailed in Appendix C.2).
>
> 3. The engineering domain follows a similar process: It is based on a real-life algorithm for fault detection in transmission lines. We converted this algorithm into a causal model and created the associated prompts (details in Appendix C.3).
>
> 4. Lastly, the math benchmarking environment is based on a particular problem within the GSM8K database. However, the original problem does not involve any interventions. We adapted the text of the problem to allow some of its story elements to be intervened on (sample prompts can be found in Appendix C.4).
>
> We are planning to release all these problems along with our code upon acceptance.
>
> ---
>
> **(3) Relationship between Reasoning and Generalization**
>
> Generalizing from a few examples to a larger input space, potentially involving never-seen-before inputs as in the case of 2-4 digit multiplication to 5-7 digit multiplication, is a useful and non-trivial property for a language model to have. This type of generalization remains a challenge within what we term the ‘in-domain’ problem. In fact, the results in Figure 5 show how DPO+CCF outperforms other baselines in this regard.
>
> While there are many notions of generalization, what we argue is that *reasoning* fundamentally involves breaking a problem down to its basic components and synthesizing them in novel ways that enable generalization to entirely new problems. In a causal graph, this may correspond to an understanding of individual edge relations in a way that would enable one to make estimations for any composite relationship. This is the type of generalization we are interested in. Here, sample efficiency in learning to solve a fixed problem would not necessarily be indicative of reasoning ability.
>
> **Update:** We have now clarified that our focus is on a specific type of generalization that aims to isolate reasoning, rather than alternative notions of generalization such as sample efficiency (Footnote 3 and the newly added Appendix E in the revised version).

---

> > ### Comment · Reviewer_fqJm · 2024-11-25
> >
> > You say "While the reasoning problems we consider are straightforward to solve once translated into a formal representation (like the structural equations in Appendix C), extracting such formal representations from the natural language descriptions of the problems remains a significant bottleneck for performance. "
> >
> > The second part of the above ("extracting such formal representations from the natural language descriptions of the problems remains a significant bottleneck for performance.") may  be true in general. But, based on the examples, such as in Figure 5, I am not sure this is true with respect to the dataset that you have created for your paper. Could you provide some evidence with respect to your dataset; not about GSM8K or other datasets. I think a few shot prompt may quite accurately extract the formal representation. Have you tried that?

---

> ### Author Response · Authors · 2024-11-26
> **Re: Official Comment by Reviewer fqJm (Comment 1/2)**
>
> The “Math Benchmarking” domain in our paper is based on the GSM8K dataset, which is why we referenced OpenMathInstruct-1 in our response; their results are relevant to our setting.
>
> That being said, your suggestion to evaluate our model of interest, Phi-3-Mini, directly on our own reasoning problems in terms of its ability to extract formal representations is an excellent one. For this purpose, we designed the following one-shot prompt using the problem in Figure 5 as an example. We then used this prompt to extract structural equations for our other problem domains: Healthcare, Engineering, and Math Benchmarking. The results are detailed further below.
>
> Prompt:
> >I want you to generate structural equations from a causal scenario described in English. Respond only with the structural equations. Here is an example:\
> >\
> >Scenario:\
> >Anna, Bill, Cory, and Dave are going to a party, where the host is going to distribute candies. Anna will be happy if she gets at least 4 candies. Bill will be happy if he gets at least 6 candies. Cory will be happy if Anna and Bill are both happy or if he gets at least 8 candies. Dave will be happy if Anna and Bill are both happy or if he gets at least 10 candies.\
> >\
> >Structural Equations:\
> >NA: The number of candies Anna gets\
> >NB: The number of candies Bill gets\
> >NC: The number of candies Cory gets\
> >ND: The number of candies Dave gets\
> >HA: Happiness of Anna\
> >HB: Happiness of Bill\
> >HC: Happiness of Cory\
> >HD: Happiness of Dave\
> >HA = (NA >= 4)\
> >HB = (NB >= 6)\
> >HC = (HA and HB) or (NC >= 8)\
> >HD = (HA and HB) or (ND >= 10)\
> >\
> >Scenario:\
> >{Test Scenario}\
> >\
> >Structural Equations:
>
> ---
>
> **Results for Healthcare**
>
> Scenario:
> > There are four types of breast cancer patients (based on their ERPR and HER2 indicators): (1) If a patient is ERPR positive and HER2 negative, they are 'Luminal A'. All luminal A patients should undergo surgery. (2) If a patient is ERPR positive and HER2 positive, they are 'Luminal B'. Luminal B patients should undergo surgery if their tumor is smaller than 1 cm and there is no nodal involvement. Luminal B patients should undergo therapy if their tumor is larger than 1 cm or if there is nodal involvement. (3) If a patient is ERPR negative and HER2 positive, they are 'Enriched'. Enriched patients should undergo surgery if their tumor is smaller than 1 cm and there is no nodal involvement. Enriched patients should undergo therapy only if their tumor is larger than 1 cm (even if there is nodal involvement). (4) If a patient is ERPR negative and HER2 negative, they are 'Basal'. Basal patients should undergo surgery if their tumor is smaller than 1 cm and there is no nodal involvement. Basal patients should undergo therapy only if their tumor is larger than 1 cm (even if there is nodal involvement).
>
> Response of Phi-3-Mini:
> >NA: ERPR status of the patient\
> >NH: HER2 status of the patient\
> >TA: Tumor size of the patient\
> >TN: Nodal involvement of the patient\
> >SA: Surgery for the patient\
> >TH: Therapy for the patient\
> >LA: Luminal A patient\
> >LB: Luminal B patient\
> >EN: Enriched patient\
> >B: Basal patient\
> >SA = (NA = 'ERPR positive' and NH = 'HER2 negative' and TA < 1 cm and TN = 'No nodal involvement')\
> >LB = (NA = 'ERPR positive' and NH = 'HER2 positive' and (TA < 1 cm and TN = 'No nodal involvement' or TA > 1 cm or TN = 'Nodal involvement'))\
> >EN = (NA = 'ERPR negative' and NH = 'HER2 positive' and (TA < 1 cm and TN = 'No nodal involvement' or TA > 1 cm))\
> >B = (NA = 'ERPR negative' and NH = 'HER2 negative' and (TA < 1 cm and TN = 'No nodal involvement' or TA > 1 cm))\
> >LA = (NA = 'ERPR positive' and NH = 'HER2 negative')\
> >LB = (NA = 'ERPR positive' and NH = 'HER2 positive')\
> >EN = (NA = 'ERPR negative' and NH = 'HER2 positive')\
> >B = (NA = 'ERPR negative' and NH = 'HER2 negative')\
> >SA = (LA or LB or EN or B)\
> >TH = (not SA)
>
> Beyond whether it is accurate or not, this response lacks a clear formal interpretation, as many of the variables are defined multiple times (causing ambiguity).

---

> ### Author Response · Authors · 2024-11-26
> **Re: Official Comment by Reviewer fqJm (Comment 2/2)**
>
> **Results for Engineering**
>
> Scenario:
> > The type of fault on a transmission line is determined through three factors X, Y, and Z. These factors are 'close to zero' if they are less than 0.1. (1) If only one of the factors is close to zero, it is a line-to-line fault. When there is a line-to-line fault, it is BC fault if factor X is close to zero, AC fault if factor Y is close to zero, and AB fault if factor Z is close to zero. (2) If exactly two of the factors are close to zero, it is a line-to-ground fault. When there is a line-to-ground fault, it is AG fault if factors Y and Z are both close to zero, BG fault if factors X and Z are both close to zero, and CG fault if factors X and Y are both close to zero.
>
> Response of Phi-3-Mini:
> > XC = (X < 0.1) and (Y >= 0.1) and (Z >= 0.1)\
> > YC = (X >= 0.1) and (Y < 0.1) and (Z >= 0.1)\
> > ZC = (X >= 0.1) and (Y >= 0.1) and (Z < 0.1)\
> > BC_fault = XC\
> > AC_fault = YC\
> > AB_fault = ZC\
> > YZ_close = (Y < 0.1) and (Z < 0.1)\
> > XZ_close = (X < 0.1) and (Z < 0.1)\
> > XY_close = (X < 0.1) and (Y < 0.1)\
> > AG_fault = YZ_close\
> > BG_fault = XZ_close\
> > CG_fault = XY_close
>
> This response is accurate except that it marginalizes out the general fault types, namely line-to-line and line-to-ground. While it is possible to compute factual values for specific fault types using these equations, the same is not true for the general fault types. Additionally, this limitation prevents us from performing any interventions on the general fault types.
>
> ---
>
> **Results for Math Benchmarking (A Modified GSM8K Problem)**
>
> Scenario:
> > Carla is downloading a 250 GB file. Normally she can download 2 GB/minute, but in 100 minutes, Windows will force a restart to install updates, which takes 20 minutes. After the restart, Carla can resume her download. Consider how long it will take to download the file.
>
> Response of Phi-3-Mini:
> > NA: The amount of data Carla downloads before the restart\
> > NR: The amount of data Carla downloads after the restart\
> > T: Total time to download the file\
> > NA = 2 GB/minute * 100 minutes\
> > NR = (250 GB - NA) / 2 GB/minute\
> > T = 100 minutes + 20 minutes + NR/2 minutes
>
> This response underlines the additional challenges involved in extracting *causal* equations. First of all, the calculations provided in the response are not entirely accurate, even for the specific instance of the scenario presented. Setting aside these small inaccuracies, the response assumes that a restart will necessarily occur. With these equations, it is not possible to reason about outcomes under interventions such as “What if Windows had not forced a restart?” or “What if Carla had been downloading a file half the size?” — both of which are types of interventions considered in our experiments.
>
> Even if we were to modify these equations to explicitly include whether a restart occurs or not as a variable, it may not always be possible to anticipate what other kinds of interventions will be queried. For instance, these equations also fail to account for potential interventions on the download speed.
>
> ---
>
> We sincerely thank you for engaging deeply with both our work and our rebuttal, and for highlighting this alternative approach to eliciting reasoning. While the equations extracted using our one-shot prompt may not be sufficient to support causal reasoning, the result for the Engineering domain was promising. Certainly, further research could explore improving the accuracy of these equations through prompt engineering or fine-tuning.
>
> Most importantly, these responses definitely helped us better understand the challenges associated with general causal reasoning, especially the response for Math Benchmarking. Accordingly, we will include these interactions with Phi-3-Mini as part of the discussion in Appendix D.2.

---

> > ### Comment · Reviewer_fqJm · 2024-11-27
> > **When should we pursue a two step approach and when to use LLM end-to-end**
> >
> > Thank you for following up on my suggestions.
> >
> > A question that remains in my head is: When should we pursue a two step approach (translate the NL description to a formal representation using LLMs  and then use a specialized solver) and when to use LLMs (or neural models) end-to-end?
> >
> > If we have a specialized solver which is accurate the two step approach boils down "translate the NL description to a formal representation".
> >
> > In case of GSM data, the tasks are grade school math word problems that is covered in countless text that are part of the LLM pre-training corpus. Thus it is reasonable to expect that LLMs do them end-to-end. Similarly about some simple logical reasoning problems. But for counterfactual reasoning: (i) Is it reasonable to expect that LLMs be able to do it? (ii) Will doing it by LLMs end to end be more accurate than using a two step approach?
> >
> > I am not convinced that doing it by LLMs end to end would be more accurate than using a two step approach, when the second step is well defined and can be coded easily.

---

> > > ### Author Response · Authors · 2024-11-29
> > > **Re: When should we pursue a two step approach and when to use LLM end-to-end**
> > >
> > > While the questions you have raised are excellent research questions, they fall outside the scope of our paper. Our primary goal is to see whether the end-to-end reasoning ability of a language model can be improved. Answering this question required us to carefully define appropriate measures of reasoning ability (through inconsistency rates as well as different modes of generalization). A key finding of our work is the importance of counterfactual feedback in improving reasoning.
> > >
> > > It is important to clarify that our objective was *not* to develop a system that can accurately estimate counterfactuals for the type of problems used in our experiments. Rather, these problems are simply a means to measure reasoning ability. To draw an analogy, consider a researcher aiming to improve the mathematical capabilities of LLMs. Their goal might be to ensure the models produce answers that are mathematically consistent. They might use GSM8K as a testbed to evaluate their methods. In doing so, their goal is not to solve GSM8K problems or to build a system capable of solving GSM8K-like problems (even if that system involves an orthogonal use of LLMs, such as translating math problems into code). In a similar vein, we want to train LLMs that produce answers with sound reasoning, not engineer a methodology to solve reasoning problems.
> > >
> > > Therefore, the relevance of a two-step approach to *our* research question, whether the reasoning ability of an LLM can be improved, is limited. *Can the two step approach lead to better counterfactual estimates in our example problems?* Based on the current state of Phi-3-Mini, the answer is clearly no. *Could the two-step approach possibly lead to better counterfactual estimates if the initial translation step is better engineered?* Maybe, we do not really know. However, this is not the goal of our paper. Again, our goal is to improve the end-to-end reasoning ability of a target language model. We define that ability as having counterfactual consistency in a language model’s answers, which is a requirement for sound reasoning about necessity/sufficiency.
> > >
> > > On a more technical level, note that the output of the language model, $\ell$, remains a natural-language answer, denoted as $a$. To evaluate the "reasoning quality" of these answers, we extract the binary outcomes expressed within them as $\hat{y} = h(a)$ and calculate various inconsistency rates that we introduced. **Our objective is not to obtain accurate $\hat{y}$ but to ensure answers $a$ exhibit good reasoning** (retaining natural-language elements such as explanations of how the final outcome was reached).

---

### Official Review · Reviewer_uHCM · 2024-11-04

**Soundness:** 3
**Presentation:** 3
**Contribution:** 3
**Rating:** 6
**Confidence:** 3

**Summary:**

This paper studies causal reasoning in LLMs through counterfactual question answering.

The authors propose a number of metrics: necessity inconsistency rate (N-IR), sufficiency inconsistency rate (S-IR), absent necessity inconsistency rate (AN-IR), absent sufficiency inconsistency rate (AS-IR), and the average of all those (Avg-IR). These go beyond simple accuracy by assessing the consistency between factual and counterfactual answers within a causal framework.

They perform a number of experiments where they finetune a small LLM (Phi-3 mini, 3.8B parameters) with SFT and DPO and present results on the extent to which their fine-tuning methods effect different generalization modes (common-cause, common-effect, inductive, deductive).

**Strengths:**

* Introducing more finegrained metrics: The proposed metrics for causal consistency are a valuable contribution, addressing a limitation of existing evaluation methods that focus primarily on accuracy.
* Generalization modes:  The proposed classes for generalization modes provide a structured framework for evaluating reasoning transfer.
* Experiments:  The paper includes a comprehensive set of experiments, including a hand-crafted puzzle and real-world problems.

**Weaknesses:**

* The choice of a very small LLM (Phi-3 mini) with limited reasoning capabilities makes it difficult to draw any conclusions for stronger models that are more commonly used for reasoning tasks.
* Clarity and Presentation:  While the paper focuses on an interesting problem and makes interesting suggestions, the presentation could be significantly improved. I found the formal definitions and descriptions of the methods a bit difficult to follow.
* Analysis of generalization: The analysis of the generalization results could be more in-depth.
* Comparison to CoT: The authors claim (in a footnote) that the effectiveness of CoT "can be attributed to
successful imitation of the provided examples". It would have been interesting to test that claim with the analytic tools they developed.

**Questions:**

1. Could you provide a more intuitive explanation of the inconsistency rate metrics in the paper?
2. Can you include the specific architecture and number of parameters of the Phi-3 mini model and discuss the limitations introduced by this choice? Can you include the Phi-3 mini performance on common reasoning benchmarks (compared to what top models are capable of)?
3. Have you considered comparing your methods with chain-of-thought prompting (see weakness above)?

---

> ### Author Response · Authors · 2024-11-22
> **Response to Reviewer uHCM (Comment 1/2)**
>
> Thank you for your thoughtful comments and suggestions! We give answers to each in turn, as well as pointing out corresponding updates we have made to the paper (changes are highlighted in blue in the revised paper).
>
> ---
>
> **(1) Choice of Phi-3-Mini**
>
> We chose a small language model like Phi-3-Mini, with limited reasoning capabilities, because it offers significant room for improvement — making it ideal for demonstrating the effectiveness of our methodology as well as highlighting the differences between various approaches more clearly.
>
> ---
>
> **(2) Architecture and Reasoning Performance of Phi-3-Mini**
>
> In all our experiments, we specifically used the version of Phi-3-Mini available on https://huggingface.co/microsoft/Phi-3-mini-128k-instruct. This version has a context length of 128k and 3.8B parameters, and has been fine-tuned for instruction following. Below is a summary of its performance on various reasoning benchmarks compared to other language models (copied from the link above). Overall, Phi-3-Mini demonstrates competitive performance.
>
> | Benchmark | Phi-3-Mini-128K-Ins | Gemma-7B | Mistral-7B | Mixtral-8x7B | Llama-3-8B-Ins | GPT3.5-Turbo-1106 |
> |---|---|---|---|---|---|---|
> | ARC Challenge  10-shot | 85.5 | 78.3 | 78.6 | 87.3 | 82.8 | 87.4 |
> | BoolQ  0-shot | 77.1 | 66 | 72.2 | 76.6 | 80.9 | 79.1 |
> | MedQA  2-shot | 56.4 | 49.6 | 50 | 62.2 | 60.5 | 63.4 |
> | OpenBookQA  10-shot | 78.8 | 78.6 | 79.8 | 85.8 | 82.6 | 86 |
> | PIQA  5-shot | 80.1 | 78.1 | 77.7 | 86 | 75.7 | 86.6 |
> | GPQA  0-shot | 29.7 | 2.9 | 15 | 6.9 | 32.4 | 29.9 |
> | Social IQA  5-shot | 74.7 | 65.5 | 74.6 | 75.9 | 73.9 | 68.3 |
> | TruthfulQA (MC2)  10-shot | 64.8 | 52.1 | 53 | 60.1 | 63.2 | 67.7 |
> | WinoGrande  5-shot | 71.0 | 55.6 | 54.2 | 62 | 65 | 68.8 |
>
> **Update:** We have now included this information in Appendix C.
>
> ---
>
> **(3) Intuitive Explanation of Inconsistency Rates**
>
> An alternative interpretation of the inconsistency rates we introduce is to frame reasoning about necessity or sufficiency as a classification problem. For instance, focusing on necessity, each context variable $U$, along with the cause $X$ and the potential effect $Y,Y_{X’}$ induced by $U$, needs to be assigned to one of three classes:
>
> 1. $\mathbb{N}$: The cause and the effect both occurred together, and the cause was necessary for the effect to occur, meaning the effect would not have occurred otherwise if the cause was prevented. In a case like this, one might say “For the observed effect, the cause was a necessary condition.”
>
> 2. $\mathbb{N}’$: The cause and the effect both occurred together, but the cause was not necessary for the effect to occur, meaning the effect would have occurred anyways even if the cause was prevented. In a case like this, one might say “For the observed effect, the cause was not a necessary condition.”
>
> 3. $\emptyset$: It is not meaningful to talk about necessity as either the cause or the effect did not occur. Note that the previous cases take the occurrence of the cause and the effect as a given, and make statements about what would have happened to the effect if the cause were to be prevented. This is consistent with Pearl’s definition of PN as a conditional probability with the condition being $X=x$ and $Y=y$.
>
> According to the underlying causal model, each context has a corresponding ground-truth “label” as defined in Equation 6. Given a language model, its answers provide estimates for the unknown potential outcomes, which in turn imply a prediction of this ground-truth “label”. The necessity inconsistency rate (N-IR) is the error rate for this classification task (how often the predicted label differs from the ground-truth label).
>
> **Update:** We have now included this alternative explanation of the inconsistency rate metrics in the newly added Appendix F.
>
> ---
>
> **(4) Detailed Analysis of Generalization Results**
>
> **Update:** We have now included a more detailed analysis of the generalization results  in the newly added Appendix G. There, we not only expand upon the findings presented in the main paper (in “Results”, Section 6.2, now 5.2 in the revised version) but also highlight additional observations. In particular, we explain why cause-based deduction, under no direct effect (“NDE”), results in larger improvement in S-IR while effect-based deduction leads to larger improvement in N-IR. In short, this is a consequence of $A$ being sufficient for $B$ and $B$ being sufficient for $C$ in our constructed example, illustrating how the underlying problem structure can influence the effectiveness of different modes of reasoning generalization.

---

> ### Author Response · Authors · 2024-11-22
> **Response to Reviewer uHCM (Comment 2/2)**
>
> **(5) Relationship to Chain-of-Thought Prompting**
>
> Responses of a language model can be improved in different ways, two of which are fine-tuning and prompt engineering. When eliciting reasoning, our focus has been fine-tuning while chain-of-thought prompting is a form of prompt engineering. These two approaches mainly differ in terms of what point they intervene on: Prompt engineering modifies the inputs passed to the language model whereas fine-tuning modifies the language model itself. Importantly, this means that the use of these two techniques are largely orthogonal to each other and definitely not mutually exclusive. In our case, chain-of-thought prompting is not a competing method against ours but rather a separate avenue for further improvement that could be pursued in conjunction with fine-tuning. Hence why we did not consider it as a baseline in our experiments.
>
> Given this context, the purpose of our footnote was to highlight that chain-of-thought prompting is subject to the same evaluation challenge that our fine-tuning approach faces: Examples provided in a chain-of-thought prompt might lead to more accurate answers for “in-domain” questions, however, this does not mean the examples was successful in eliciting reasoning unless the accuracy generalizes to new problems (that require forming novel chains using the same concepts).
>
> **Update:** We have now included this more detailed discussion in a newly added Appendix D.1. We have also clarified the corresponding footnote.

---

> > ### Comment · Reviewer_uHCM · 2024-11-24
> >
> > Thank you for the detailed response, and the significant improvements to the paper addressing some of my concerns. I am increasing my score.
> >
> > There are several good reasons to choose smaller models, but I don't believe the one you mentioned makes sense. Larger models will likely offer more significant room for improvements.

---

> > > ### Author Response · Authors · 2024-11-26
> > > **Re: Official Comment by Reviewer uHCM**
> > >
> > > Thank you for engaging with our response and updating your score!
> > >
> > > **Regarding small language models offering more significant room for improvement,** the initial error rate of a language model determines the largest possible improvement we can observe during our experiments. At one extreme, if the model already makes no errors, then there is no room for improvement. Conversely, when the model has a higher initial error rate, the potential margins of improvement are larger, making it easier to compare different methods and settings with each other.
> > >
> > > For example, consider the example we give in Figure 1, where Phi-3-Mini has a factual ER of ~15% and a counterfactual ER of ~30%. In contrast, GPT-4, performing the same task, makes practically no factual errors and has a counterfactual ER of only ~2.5% (see Figure 1 in Gonzalez & Nori, 2024, https://arxiv.org/pdf/2408.08210). During our experiments, if we had used a language model with near-zero factual error, we would never have observed a significant improvement in F-ER, and therefore, would not have been able to make the observation that common-effect generalization tends to improve F-ER more than CF-ER. (Varying causes has no impact on factual questions regarding a fixed effect whereas counterfactual questions change. In Figure 6a, this asymmetry leads to a greater improvement in N-IR than in S-IR, because in our constructed problem where causes are largely sufficient for effects, determining necessity rarely requires estimating counterfactuals).

---

### Meta-Review · Area_Chair_cyS4 · 2024-12-19

**Metareview:**

This paper is truly in the borderline with varied reviews. The reviewers liked the proposed metrics for causal consistency in the counterfactual question answering, strong empirical results, and the general positioning of the paper.

The key weaknesses were in the terms of examples that can be improved to match the motivations and experiments, analysis of generalization results and restriction of experiments (even though the results are strong) to one model and missing experimental details.

When I read the paper, i share many of the concerns but the paper presents an exciting problem and the issues can be addressed with minor revisions. So I recommend acceptance.

**Additional Comments On Reviewer Discussion:**

There was not much discussion between the reviewers but they interacted with the authors. On reading the discussions and the responses from the authors, it is clear to me that the authors have earnestly and politely addressed many of the concerns. So it is clear to me that the paper will benefit from presenting at ICLR. It is an important contribution to LLMs and counterfactual reasoning.

---

### Decision · Program_Chairs · 2025-01-22

Accept (Oral)